# Posterior Inference on Shallow Infinitely Wide Bayesian Neural Networks under Weights with Unbounded Variance

**Jorge Loría**[1,2]          **Anindya Bhadra**[1]

[1]Department of Statistics, Purdue University, West Lafayette, Indiana, USA
[2]Department of Computer Science, Aalto University, Finland

## Abstract

From the classical and influential works of Neal (1996), it is known that the infinite width scaling limit of a Bayesian neural network with one hidden layer is a Gaussian process, *when the network weights have bounded prior variance*. Neal's result has been extended to networks with multiple hidden layers and to convolutional neural networks, also with Gaussian process scaling limits. The tractable properties of Gaussian processes then allow straightforward posterior inference and uncertainty quantification, considerably simplifying the study of the limit process compared to a network of finite width. Neural network weights with unbounded variance, however, pose unique challenges. In this case, the classical central limit theorem breaks down and it is well known that the scaling limit is an $\alpha$-stable process under suitable conditions. However, current literature is primarily limited to forward simulations under these processes and the problem of posterior inference under such a scaling limit remains largely unaddressed, unlike in the Gaussian process case. To this end, our contribution is an interpretable and computationally feasible procedure for posterior inference, using a *conditionally Gaussian* representation, that then allows full use of the Gaussian process machinery for tractable posterior inference and uncertainty quantification in the non-Gaussian regime.

## 1 INTRODUCTION

Gaussian processes (GPs) have been studied as the infinite width limit of Bayesian neural networks with priors on network weights that have finite variance (Neal, 1996; Williams, 1996). This presents some key advantages over

Bayesian neural networks with finite widths that usually require computation intensive Markov chain Monte Carlo (MCMC) posterior calculations (Neal, 1996) or variational approximations (Goodfellow et al., 2016, Chapter 19); in contrast to straightforward posterior inference and probabilistic uncertainty quantification afforded by the GP machinery (Williams and Rasmussen, 2006). In this sense, the work of Neal (1996) is foundational. The technical reason for this convergence to a GP is due to an application of the central limit theorem under the bounded second moment condition. More specifically, given an $I$ dimensional input $\mathbf{x}$ and a one-dimensional output $y(\mathbf{x})$, a $K$ layer feedforward deep neural network (DNN) with $K - 1$ hidden layers is defined by the recursion:

$$z_j^{(l+1)}(\mathbf{x}) = g\left(b_j^{(l)} + \sum_{i=1}^{p_l} w_{ij}^{(l)} z_i^{(l)}(\mathbf{x})\right), \quad l = 1, \ldots, K - 1, \tag{1}$$

$$y(\mathbf{x}) = \sum_{j=1}^{p_K} w_j^{(K)} z_j^{(K)}(\mathbf{x}), \tag{2}$$

where $z^{(1)} \equiv \mathbf{x}$, $p_1 = I$, $p_K = D$ and $g(\cdot)$ is a nonlinear activation function. Thus, the network repeatedly applies a linear transformation to the inputs at each layer, before passing it through a nonlinear activation function. Sometimes a nonlinear transformation is also applied to the final hidden layer to the output layer, but in this paper it is assumed the output is a linear function of the last hidden layer. Neal (1996) considers the case of a Bayesian neural network with a single hidden layer, i.e., $K = 2$. So long as the hidden to output weights $w^{(2)}$ are independent and identically distributed Gaussian, or at least, have a common bounded variance given by $c/p_2$ for some $c > 0$, and $g(\cdot)$ is bounded, an application of the classical central limit theorem shows the network converges to a GP as the number of hidden nodes $p_2 \to \infty$.

### 1.1 RELATED WORKS

The foundational work of Neal (1996) was followed by an explicit computation of some of the kernels obtained from

this limiting process (Williams, 1996). Recently, Neal's result has been extended to prove fully connected multi-layer feedforward networks (Lee et al., 2018; de G. Matthews et al., 2018) and convolutional neural networks (Garriga-Alonso et al., 2019; Novak et al., 2019) also converge to GPs. The Tensor Program of Yang (2019) has successfully extended these results to feedforward and recurrent networks of *any architecture.* This is useful for uncertainty quantification by designing emulators for deep neural networks (DNNs) based on GPs, since the behavior of finite-dimensional DNNs for direct uncertainty quantification is much harder to characterize. In contrast, once a convergence to GP can be ensured, well established tools from the GP literature (see, e.g., Williams and Rasmussen, 2006) can be brought to the fore to allow straightforward posterior inference. The induced covariance function depends on the choice of the nonlinear activation function $g(\cdot)$, and is in general anisotropic. However, it can be worked out in explicit form under a variety of activation functions for both shallow (Neal, 1996; Williams, 1996; Cho and Saul, 2009) and deep (Lee et al., 2018; de G. Matthews et al., 2018) feedforward neural networks, where for deep networks usually a recursive formula is available that expresses the covariance function of a given layer conditional on the previous layer. The benefit of depth is that it allows a potentially very rich covariance function at the level of the observed data, even if the covariances in each layer conditional on the layer below are simple. Viewing a GP as a prior on the function space, this allows for a rich class of prior structures. However, the process is still Gaussian in all these cases and our intention in this paper is a departure from the Gaussian world.

For finite width neural networks, non-Gaussian weights have recently been considered by Fortuin et al. (2022) and Fortuin (2022). Departures from $i.i.d.$ weights have also recently received attention (Caron et al., 2023; Lee et al., 2023). Theoretical results with infinite variance were hinted at by Neal (1996), and first proved by Der and Lee (2005). Follow-up theoretical results have been obtained in varied architectures for bounded activation functions (Peluchetti et al., 2020; Bracale et al., 2022) and with unbounded activation functions (Bordino et al., 2022). However, posterior inference still remains challenging in the infinite-width limit, due to the reasons made clear in the next sub-section.

## 1.2 CHALLENGES POSED BY NETWORK WEIGHTS WITH UNBOUNDED PRIOR VARIANCE

Although the GP literature has been immensely influential for uncertainty quantification in DNNs, it is obvious that a DNN does not converge to a GP if the final hidden to output layer weights are allowed to have unbounded variance, e.g., belonging to $t$ or others in the stable family, such that the scaling limit distribution is non-Gaussian (Gnedenko and Kolmogorov, 1954). This was already observed by Neal (1996) who admits: *"in contrast to the situation for Gaussian process priors, whose properties are captured by their covariance functions, I know of no simple way to characterize the distributions over functions produced by the priors based on non-Gaussian stable distributions."* Faced with this difficulty, Neal (1996) confines himself to forward simulations from DNNs with $t$ weights, and yet, observes that the network realizations under these weights demonstrate very different behavior (e.g., large jumps) compared to normal priors on the weights. This is not surprising, since Gaussian processes, with their almost surely continuous sample paths, are not necessarily good candidate models for functions containing sharp jumps, perhaps explaining their lack of popularity in certain application domains, e.g., finance, where jumps and changepoints need to be modeled (see, e.g., Chapter 7 of Cont and Tankov, 2004). Another key benefit of priors with polynomial tails, pointed out by Neal (1996), is that it allows a few hidden nodes to make a large contribution to the output, while drowning out the others, akin to feature selection. In contrast, in the GP limit, the contributions of individual nodes are averaged out. Thus, there are clear motivations for developing computationally feasible posterior inference machinery under these non-Gaussian limits.

Neal (1996) further hints that it may be possible to prove an analogous result using priors that do not have finite variance. Specifically, suppose the network weights are given symmetric $\alpha$-stable priors, which have unbounded variance for all $\alpha \in (0, 2)$, and the $\alpha = 2$ case coincides with a Gaussian random variable. If $X$ is an $\alpha$-stable random variable, the density does not in general have a closed form, but the characteristic function is:

$$\phi_X(t) = \exp[it\mu - \nu^\alpha |t|^\alpha \{1 - i\beta \text{sign}(t)\omega(t; \alpha)\}],$$

where $\omega(t; \alpha) = \tan(\alpha\pi/2)$, for $\alpha \neq 1$ and $\omega(t; \alpha) = -(2/\pi)\log|t|$, for $\alpha = 1$. Here $\mu \in \mathbb{R}$ is called the shift parameter, $\alpha \in (0, 2]$ is the index parameter, $\beta \in [-1, 1]$ is the symmetry parameter, and $\nu > 0$ is the scale parameter (Samorodnitsky and Taqqu, 1994, p. 5). Throughout, we use a zero shift ($\mu = 0$) stable variable, and denote it by $X \sim S(\alpha, \nu, \beta)$. Here $\beta = 0$ corresponds to the symmetric case, and when $\beta = 1, \alpha < 1, \nu = 1$, the random variable is strictly positive, which we denote by $S^+(\alpha)$. We refer the reader to Supplementary Material S.1 for some further properties of $\alpha$-stable random variables, as relevant for the present work. Der and Lee (2005) confirm Neal's conjecture by establishing that the scaling limit of a shallow neural network under $\alpha$-stable priors on the weights is an $\alpha$-stable process. Proceeding further, Peluchetti et al. (2020) show that the limit process for infinitely wide DNNs with infinite-variance priors is also an $\alpha$-stable processes. However, both Der and Lee (2005) and Peluchetti et al. (2020) only consider the forward process and neither considers posterior inference. Inference using $\alpha$-stable densities is not straight-

forward, and some relevant studies are by Samorodnitsky and Taqqu (1994), Lemke et al. (2015), and more recently by Nolan (2020). The main challenge is that a covariance function is not necessarily defined, precluding posterior inference analogous to the GP case, for example, using the *kriging* (Stein, 1999) machinery. To this end, our contribution lies in using a representation of the characteristic function of symmetric $\alpha$-stable variables as a normal scale mixture, that then allows a *conditionally Gaussian* representation. This makes it possible to develop posterior inference and prediction techniques under stable priors on network weights using a latent Gaussian framework.

### 1.3 SUMMARY OF MAIN CONTRIBUTIONS

Our main contributions consist of:

1. An explicit characterization of the posterior predictive density function under infinite width scaling limits for shallow (one hidden layer) Bayesian neural networks under stable priors on the network weights, using a latent Gaussian representation.

2. An MCMC algorithm for posterior inference and prediction, with publicly available code.

3. Numerical experiments in one and two dimensions that validate our procedure by obtaining better posterior predictive properties for functions with jumps and discontinuities, compared to both Gaussian processes and Bayesian neural networks of finite width.

4. A real world application on a benchmark real estate data set from the UCI repository.

## 2 INFINITE WIDTH LIMITS OF BAYESIAN NEURAL NETWORKS UNDER WEIGHTS WITH UNBOUNDED VARIANCE

Consider the case of a shallow, one hidden layer network, with the weights of the last layer being independent and identically distributed with symmetric $\alpha$-stable priors. Our results are derived under this setting using the following proposition of Der and Lee (2005).

**Proposition 1.** *(Der and Lee, 2005). Let the network specified by Equations* (1) *and* (2)*, with a single hidden layer* ($K = 2$)*, have i.i.d. hidden-to-output weights* $w_j^{(2)}$ *distributed as a symmetric $\alpha$-stable with scale parameter* $(\nu/2)^{1/2}p_2^{-1/\alpha}$. *Then* $y(\mathbf{x})$ *converges in distribution to a symmetric $\alpha$-stable process* $f(\mathbf{x})$ *as* $p_2 \to \infty$ *for random input-to-hidden weights. The finite dimensional distribution of* $f(\mathbf{x})$*, denoted as* $(f(\mathbf{x}_1), \ldots, f(\mathbf{x}_n))$ *for all* $n$*, where* $\mathbf{x}_i \in \mathbb{R}^I$*, is multivariate stable with a characteristic function:*

$$\phi(\mathbf{t}) = \mathbb{E}\left[\exp\{i\langle \mathbf{t}, f(\mathbf{x})\rangle\}\right]$$

$$= \exp\left\{-(\nu/2)^{\alpha/2}\mathbb{E}[|\langle \mathbf{t}, \mathbf{g}\rangle|^{\alpha}]\right\}, \qquad (3)$$

*where angle brackets denote the inner product,* $\mathbf{t} = (t_1, \ldots, t_n)$ *is the argument of the characteristic function,* $\mathbf{g} = (g(\mathbf{x}_1), \ldots, g(\mathbf{x}_n))$*, and* $g(\mathbf{x})$ *is a random variable with the common distribution (across $j$) of* $(z_j^{(2)}(\mathbf{x}_1), \ldots, z_j^{(2)}(\mathbf{x}_n))$.

Following Neal (1996), assume for the rest of the paper that the activation function $g(\cdot)$ corresponds to the sign function: $\text{sign}(\xi) = 1$, if $\xi > 0$; $\text{sign}(\xi) = -1$, if $\xi < 0$; and $\text{sign}(0) = 0$. For $\xi \in \mathbb{R}^I$ we define $g(\xi) = \text{sign}\left(b_0 + \sum_{i=1}^{I} w_i\xi_i\right)$, where $b_0$ and $w_i$ are i.i.d. standard Gaussian variables. The next challenge is to compute the expectation within the exponential in Equation (3). To resolve this, we break it into simpler cases. Define $\Lambda$ as the set of all possible functions $\tau : \{\mathbf{x}_1, \ldots, \mathbf{x}_n\} \to \{-1, +1\}$. Noting that each $\mathbf{x}_j$ can be mapped to two possible options: $+1$ and $-1$, indicates that there are $2^n$ elements in $\Lambda$. For each $\ell = 1, \ldots, 2^n$, consider $\tau_\ell \in \Lambda$, the event $A_\ell = \{\tau_\ell(\mathbf{x}_j) = g(\mathbf{x}_j)\}_{j=1}^n$, and the probability $q_\ell = \mathbb{P}(A_\ell)$. By definition $\{A_\ell\}_{\ell=1}^{2^n}$ is a set of disjoint events. Next, using the definition of the expectation of discrete disjoint events we obtain:

$$\mathbb{E}[|\langle \mathbf{t}, \mathbf{g}\rangle|^{\alpha}] = \sum_{\ell=1}^{2^n} q_\ell \left|\sum_{j=1}^{n} t_j\tau_\ell(\mathbf{x}_j)\right|^{\alpha}, \qquad (4)$$

where the expectation is over input-to-hidden weights. A naïve enumeration sums over an exponential number of terms in $n$, and is impractical. However, details of the computation of $q_\ell$ and $\tau_\ell$ are given in Supplementary Section S.2, where we show how to reduce the enumeration over $2^n$ terms in Equation (4) to $L = \mathcal{O}(n^I)$ terms using the algorithm of Goodman and Pollack (1983), by identifying only those configurations with $q_\ell > 0$. This allows circumventing the exponential enumeration in $n$, resulting in a polynomial complexity algorithm, depending on the input dimension $I$. Although this computational complexity still appears rather high at a first glance, especially for high-dimensional inputs, for two or three-dimensional problems (e.g., in spatial or spatio-temporal models), the computation is both manageable and practical, and the complexity is similar to the usual GP regression.

### 2.1 A CHARACTERIZATION OF THE POSTERIOR PREDICTIVE DENSITY UNDER STABLE NETWORK WEIGHTS USING A CONDITIONALLY GAUSSIAN REPRESENTATION

While the previous section demonstrated the characteristic function in Equation (3) can be computed, the resulting density, obtained via its inverse Fourier transform, does not

necessarily have a closed form, apart from specific values of $\alpha$, such as $\alpha = 2$ (Gaussian), $\alpha = 1$ (Cauchy) or $\alpha = 0.5$ (inverse Gaussian). In this section we show that a *conditionally Gaussian* characterization of the density function is still possible for the entire domain of $\alpha \in (0, 2]$, facilitating posterior inference. First, note that the result of Der and Lee (2005) is obtained assuming that there is no intrinsic error in the observation model, i.e., they assume the observations are obtained as $y_i = f(\mathbf{x}_i)$, and the only source of randomness is the network weights. We generalize this to more realistic scenarios and consider an additive error term. That is, we consider the observation model $y_i = f(\mathbf{x}_i) + \varepsilon_i$, where the error terms $\varepsilon_i$ are independent identically distributed normal random variables with constant variance $\sigma^2$. Using the expression for the expectation in the characteristic function from Proposition 1, we derive the full probability density function, as specified in the following theorem, with a proof in Section 7.

**Theorem 1.** *For real-valued observations* $\mathbf{y} = (y_1, \ldots, y_n)$ *under the model* $y_i = f(\mathbf{x}_i) + \varepsilon_i$; *where* $\varepsilon_i \overset{i.i.d.}{\sim} \mathcal{N}(0, \sigma^2)$ *and* $f(\cdot)$ *is as specified in Proposition 1 , denote the matrix* $\mathbf{X} = [\mathbf{x}_1, \ldots, \mathbf{x}_n]^T$. *The probability density function of* $(\mathbf{y} \mid \mathbf{X})$ *is:*

$$p(\mathbf{y} \mid \mathbf{X}) = (2\pi)^{-n/2} \int_{(\mathbb{R}^+)^L} \exp\left(-\frac{1}{2}\mathbf{y}^T \mathbf{Q_s}^{-1} \mathbf{y}\right)$$
$$\times \det(\mathbf{Q_s})^{-1/2} \prod_{\ell=1}^{L} p_{S^+}(s_\ell) ds_\ell,$$

*where* $p_{S^+}$ *is the density for a positive* $\alpha/2$-*stable random variable, and* $\mathbf{Q_s}$ *is a positive definite matrix with probability one that depends on* $\mathbf{s} = \{s_\ell\}_{\ell=1}^{L}$. *Specifically,* $\mathbf{Q_s} = \sum_{\ell=1}^{L} s_\ell q_\ell^{2/\alpha} \boldsymbol{\tau}_\ell \boldsymbol{\tau}_\ell^T + \sigma^2 \mathbf{I}$, *denoting by* $\boldsymbol{\tau}_\ell \in \mathbb{R}^n$ *the vector with entries* $(\tau_\ell(\mathbf{x}_1), \ldots, \tau_\ell(\mathbf{x}_n))$.

Theorem 1 is the main machinery we need for posterior inference. We emphasize that $\mathbf{Q_s}$ is a matrix with random entries, conditional on $\{s_\ell\}_{\ell=1}^{L}$; and $\{q_\ell\}_{\ell=1}^{L}$ and $\{\tau_\ell\}_{\ell=1}^{L}$ are deterministic. Further, the input variables $\mathbf{X}$ are also deterministic. Theorem 1 implies the hierarchical Gaussian model:

$$\mathbf{y} \mid \mathbf{X}, \{s_\ell\}_{\ell=1}^{L} \sim \mathcal{N}_n(0, \mathbf{Q_s}), \quad s_\ell \overset{i.i.d.}{\sim} S^+(\alpha/2).$$

Each of the $\tau$s defines a level set for the points that lie in the $+1$ side in contraposition to those that lie in the $-1$ side. A forward simulation of this model is a weighted sum of the $\tau$s, with corresponding positive weight for those that lie closer and a negative weight for the points that lie farther. We further present the following corollaries to interpret the distribution of $\mathbf{Q_s}$.

**Corollary 1.** *The matrix* $\mathbf{Q_s}$ *in Theorem 1 is stochastic for all* $\alpha \in (0, 2)$ *and is deterministic when* $\alpha = 2$.

*Proof.* Recall, $\mathbf{Q_s} = \sum_{\ell=1}^{L} s_\ell q_\ell^{2/\alpha} \boldsymbol{\tau}_\ell \boldsymbol{\tau}_\ell^T + \sigma^2 \mathbf{I}$, $s_\ell \overset{i.i.d.}{\sim} S^+(\alpha/2)$. Thus, $s_\ell \to 1$, w.p. 1, as $\alpha \to 2$, since an $S^+(1)$ variable is a degenerate point mass at 1. $\square$

Noting the $\alpha = 2$ case is Gaussian, Corollary 1 indicates $\mathbf{Q_s}$ is stochastic in the stable limit, but deterministic in the GP limit; a key difference. The lack of representation learning in the GP limit, due to the kernel converging to a degenerate point mass, is a major criticism of the GP limit framework, see for example Aitchison et al. (2021); Yang et al. (2023). A useful implication of Corollary 1 is that the posterior of $\mathbf{Q_s} \mid \mathbf{y}$ is non-degenerate in the stable limit. Numerical results supporting this claim are in Supplementary Section S.4. Specifically, when $\alpha = 2$, the limiting process of Proposition 1 is a GP, which has been established to have a deterministic covariance kernel (Cho and Saul, 2009). When $\alpha < 2$, which is our main interest, Corollary 1 ensures that the *conditional* covariance kernel is stochastic, thereby enabling learning a degenerate posterior of this quantity given the data. This is at a contrast to the degenerate posterior in the Gaussian case, for both shallow and deep infinite-width limits of BNNs, as discussed by Aitchison et al. (2021). We further remark here that although the current work only considers shallow networks, this property of a non-degenerate posterior should still hold for deep networks under $\alpha$-stable weights. However, the challenge of relating the *conditional* covariance kernel of each layer to the layer below, analogous to the deep GP case (e.g., de G. Matthews et al., 2018), is beyond the scope of the current work.

**Corollary 2.** *The marginal distribution of the diagonal entries of the matrix* $\mathbf{Q_s}$ *is* $\sigma^2 + \nu S^+(\alpha/2)$, *where the* $\sigma^2$ *acts as a shift parameter, and the marginal distribution of the entry* $i, j$ *in the* $\mathbf{Q_s}$ *matrix is* $S(\alpha/2, \nu, 2p_{ij} - 1)$, *where* $p_{ij} = \sum_{\ell:\tau_\ell(\mathbf{x}_i)=\tau_\ell(\mathbf{x}_j)} q_\ell$, *the probability that* $\mathbf{x}_i$ *and* $\mathbf{x}_j$ *lie on the same side of the hyperplane partition. Further, the entries of* $\mathbf{Q_s}$ *are not independent.*

*Proof.* For $\{\mathbf{Q_s}\}_{ii} - \sigma^2$, we apply Property 1.2.1 of Samorodnitsky and Taqqu (1994), which we refer as the closure property, to obtain $\nu S^+(\alpha/2)$. Next for $\{\mathbf{Q_s}\}_{ij}$, we split the summation in two cases: $\tau_\ell(\mathbf{x}_i) = \tau_\ell(\mathbf{x}_j)$, and $\tau_\ell(\mathbf{x}_i) \neq \tau_\ell(\mathbf{x}_j)$. Using the closure property in the separate splits, we obtain $\{\mathbf{Q_s}\}_{ij} \sim S(\alpha/2, \nu p_{ij}^{2/\alpha}, 1) - S(\alpha/2, \nu(1 - p_{ij})^{2/\alpha}, 1)$. The result follows by applying the closure property once more. The entries of $\mathbf{Q_s}$ are not independent, as they are obtained from a linear combination of the independent variables $\{s_\ell\}_{\ell=1}^{L}$. $\square$

The value of this corollary does not lie in a numerical or computational speed-up, since the obtained marginals are not independent, but rather in the interpretability that it lends to the model. Explicitly, it indicates that when the points lie closer their conditional covariance is more likely

to be positive and when they lie farther apart the covariance is more likely to be negative (see Proposition 1.2.14 of Samorodnitsky and Taqqu, 1994).

Now that the probability model is clear, we proceed to the next problem of interest: prediction. To this end, we present the following proposition, characterizing the posterior predictive density.

**Proposition 2.** *Consider a vector of $n$ real-valued observations $\mathbf{y} = (y_1, \ldots, y_n)$, each with respective input variables $\mathbf{x}_1, \ldots, \mathbf{x}_n \in \mathbb{R}^I$, under the model $y_i = f(\mathbf{x}_i) + \varepsilon_i; \varepsilon_i \overset{i.i.d.}{\sim} \mathcal{N}(0, \sigma^2)$, and $m$ new input variable locations: $\mathbf{x}_1^*, \ldots, \mathbf{x}_m^* \in \mathbb{R}^I$, with future observations at those locations denoted by $\mathbf{y}^* = (y_1^*, \ldots, y_m^*)$. Denote the matrices $\mathbf{X} = [\mathbf{x}_1, \ldots, \mathbf{x}_n]^T$ and $\mathbf{X}^* = [\mathbf{x}_1^*, \ldots, \mathbf{x}_m^*]^T$. The posterior distribution at these new input variables satisfies the following properties:*

1. *The conditional posterior of $\mathbf{y}^* \mid \mathbf{y}, \mathbf{X}, \mathbf{X}^*, \mathbf{Q_s}$ is an $m$-dimensional Gaussian. Specifically:*

$$\mathbf{y}^* \mid \mathbf{y}, \mathbf{X}, \mathbf{X}^*, \mathbf{Q_s} \sim \mathcal{N}_m(\boldsymbol{\mu}^*, \boldsymbol{\Sigma}^*), \quad (5)$$

   *where $\boldsymbol{\mu}^* = \mathbf{Q}_{*,1:n}\mathbf{Q}_{1:n,1:n}^{-1}\mathbf{y}$, and $\boldsymbol{\Sigma}^* = \mathbf{Q}_{*,*} - \mathbf{Q}_{*,1:n}\mathbf{Q}_{1:n,1:n}^{-1}\mathbf{Q}_{1:n,*}$, using $\mathbf{Q_s}$ as previously defined for the $n + m$ input variables, and denoting by '$*$' the entries $(n+1):(n+m)$.*

2. *The posterior predictive density at the $m$ new locations conditional on the observations $\mathbf{y}$, is given by:*

$$
p(\mathbf{y}^* \mid \mathbf{y}, \mathbf{X}, \mathbf{X}^*) = \int_{(\mathbb{R}^+)^L} p(\mathbf{y}^* \mid \mathbf{y}, \mathbf{X}, \mathbf{X}^*, \mathbf{Q_s})
$$
$$
\times\, p(\mathbf{Q_s} \mid \mathbf{y}, \mathbf{X}) d\mathbf{Q_s}, \quad (6)
$$

   *where $p(\mathbf{y}^* \mid \mathbf{y}, \mathbf{X}, \mathbf{X}^*, \mathbf{Q_s})$ is the conditional posterior density of $\mathbf{y}^*$, $\mathbf{Q_s}$ is as previously described for the $n + m$ input variables, and the integral is over the values determined by $\{s_\ell\}_{\ell=1}^L$.*

*Proof.* The first is an immediate application of Theorem 1 and the conditional density of a multivariate Gaussian. The second part follows from a standard application of marginal probabilities. □

## 3  AN MCMC SAMPLER FOR THE POSTERIOR PREDICTIVE DISTRIBUTION

Dealing with $\alpha$-stable random variables includes the difficulty that the moments of the variables are only finite up to an $\alpha$ power. Specifically, for $\alpha < 2$, if $X \sim S(\alpha, \nu, \beta)$, then $\mathbb{E}[|X|^r] = \infty$ if $r \geq \alpha$, and is finite otherwise (Samorodnitsky and Taqqu, 1994, Property 1.2.16). To circumvent dealing with potentially ill-defined moments, we propose

to sample from the full posterior. For fully Bayesian inference, we assign $\sigma^2$ a half-Cauchy prior (Gelman, 2006) and iteratively sample from the posterior predictive distribution by cycling through $(\mathbf{y}^*, \mathbf{Q}, \sigma^2)$ in an MCMC scheme, as described in Algorithm 1, which has computational complexity of the order of $\mathcal{O}(T[(n + m)^I n^2 + m^3])$, where $T$ is the number of MCMC simulations used. The method of Chambers et al. (1976) is used for simulating the stable variables. An implementation of our algorithms is freely available at https://github.com/loriaJ/alphastableNNet.

---

**Algorithm 1** A Metropolis–Hastings sampler for the posterior predictive distribution

---

**Require:** Observations $\mathbf{y} \in \mathbb{R}^n$, with $I$-dimensional input variables $\mathbf{X} \in \mathbb{R}^{n \times I}$, new input variables $\mathbf{X}^* \in \mathbb{R}^{m \times I}$, and number of MCMC iterations $T$.
**Output:** Posterior predictive samples $\{\mathbf{y}_k^*\}_{k=1}^T$
    Obtain $\Lambda$ for $(\mathbf{X}, \mathbf{X}^*)$ using Algorithm S.1.
    Compute $\{q_\ell\}_{\ell=1}^L$ as described in Supplementary Section S.2.
    Initialize $\mathbf{Q_s}^{(0)}$ using independent samples of $s_\ell$ from the prior distributions.
    **for** $k = 1, \ldots, T$ **do**
        Simulate $\mathbf{Q_s}^{(k)} \mid \mathbf{y}, \mathbf{Q_s}^{(k-1)}$ using Algorithm S.2.
        Compute $\boldsymbol{\mu}_k^*$ and $\boldsymbol{\Sigma}_k^*$ using $\mathbf{Q_s}^{(k)}$ in Part 1 of Proposition 2.
        Simulate $\mathbf{y}_k^* \mid (\mathbf{y}, \mathbf{Q_s}^{(k)}) \sim \mathcal{N}_m(\boldsymbol{\mu}_k^*, \boldsymbol{\Sigma}_k^*)$.
    **end for**
    **return** $\{\mathbf{y}_k^*\}_{k=1}^T$.

---

There are two hyper-parameters in our model: $\alpha, \nu$. We propose to select them by cross validation on a grid of $(\alpha, \nu)$, and selecting the result with smallest mean absolute error (MAE). Another possible way to select $\alpha$ is by assigning a prior. The natural choice for $\alpha$ is a uniform prior on $(0, 2)$, however the update rule would need to consider the densities of the $L$ such $\alpha/2$-stable densities $p_{S^+}$, which would be computationally intensive as there is no closed form to this density apart from specific values of $\alpha$. A prior for $\nu$ could be included but a potential issue of identifiability emerges, similar to that identified for the Matérn kernel (Zhang, 2004). We leave these open for future research.

## 4  NUMERICAL EXPERIMENTS

We compare our method against the predictions obtained from three other methods. The first two are methods for Gaussian processes that correspond to the two main approaches in GP inference: maximum likelihood with a Gaussian covariance kernel (Dancik and Dorman, 2008), and an MCMC based Bayesian procedure using the Matérn kernel (Gramacy and Taddy, 2010); the third method is a two-layer Bayesian neural network (BNN) using a single hidden layer of 100 nodes with Gaussian priors, implemented in

pytorch (Paszke et al., 2019), and fitted via a variational approach. The choice of a modest number of hidden nodes is intentional, so that we are away from the infinite width GP limit, and the finite-dimensional behavior can be visualized. The respective implementations are in the R packages mlegp, tgp, and the python libraries pytorch and torchbnn. The estimates used from these methods are respectively the kriging estimate, posterior median and posterior mean. We tune our method by cross-validation over a grid of $(\alpha, \nu)$. We use point-wise posterior median as the estimate, and report the values with smallest mean absolute error (MAE) and the optimal parameters. Results on timing and additional simulations are in Supplementary Section S.4, including the posterior quantiles of $\mathbf{Q_s} \mid \mathbf{y}$, showing the posterior is non-degenerate and stochastic. This suggests learning a non-degenerate posterior $\mathbf{Q_s} \mid \mathbf{y}$ is possible, unlike in the GP limit where the kernel is degenerate (Aitchison et al., 2021; Yang et al., 2023). We use a data generating mechanism of the form $y = f(x) + \varepsilon$, where the $f$ is the true function. The overall summary is that when $f$ has at least one discontinuity, our method performs better at prediction than the competing methods, and when $f$ is continuous the proposed method performs just as well as the other methods. This provides empirical support that the assumption of *continuity* of the true function cannot be disregarded in the *universal approximation* property of neural networks (Hornik et al., 1989), and the adoption of infinite variance prior weights might be a crucial missing ingredient for successful posterior prediction when the truth is discontinuous.

## 4.1 EXPERIMENTS IN ONE DIMENSION

We consider a function with three jumps: $f(x) = 5 \times \mathbf{1}_{\{x \geq 1\}} + 5 \times \mathbf{1}_{\{-1 \leq x < 0\}}$, to which we add a Gaussian noise with $\sigma = 0.5$. We consider $x \in [-2, 2]$, with 40 equally spaced points as the training set, and 100 equally spaced points in the testing set. We display the two-panel Figure 1, showing the comparisons between the four methods. The boxplots in the left panel show that the proposed method – which we term *"Stable,"* has the smallest prediction error. The right panel shows that the BNN, the GP based fully Bayesian, and maximum likelihood methods have much smoother predictions than the Stable method. This indicates the inability of these methods to capture sharp jumps as well as the Stable method, which very clearly captures them. The Stable method obtains the smallest cross validation error for this case with $\alpha^* = 1.1$ and $\nu^* = 1$.

Figure 2 displays the uncertainty of the GP Bayes and Stable methods, for the same setting, using the 90% posterior predictive intervals. In general, the intervals are narrower for the stable case.

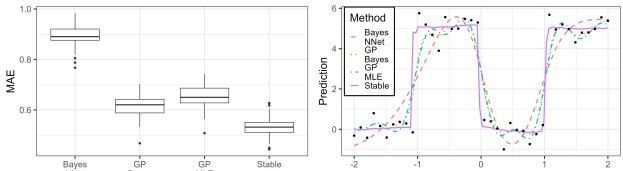

Figure 1: *Left:* Boxplots of mean absolute error of out-of-sample prediction over test points, and *Right:* predicted values over 100 points on a regular grid on $[-2, 2]$. Training points in black dots.

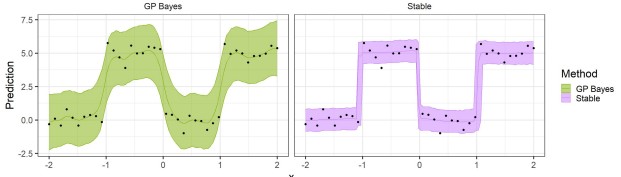

Figure 2: The point-wise 90% posterior predictive intervals for GP Bayes and Stable over 100 points on a regular grid on $[-2, 2]$, training points in black.

## 4.2 EXPERIMENTS IN TWO DIMENSIONS

We consider the function: $f(x_1, x_2) = 5 \times \mathbf{1}_{\{x_1 > 0\}} + 5 \times \mathbf{1}_{\{x_2 > 0\}}$, with additive Gaussian noise with $\sigma = 0.5$, and observations on an equally-spaced grid of 49 points in the square $[-1, 1]^2$. In Figure 3 we display the boxplots and contour plots for all methods for out-of-sample prediction on an equally spaced grid of $9 \times 9$ points in the same square. The methods that employ Gaussian processes (GP Bayes and GP MLE) and BNN seem to have smoother transitions between the different quadrants, whereas the Stable method captures the sharp jumps better. This is reinforced by the prediction errors displayed in the left panel. For this example, the Stable method obtains the smallest cross validation error with $\alpha^* = 1.1, \nu^* = 1$.

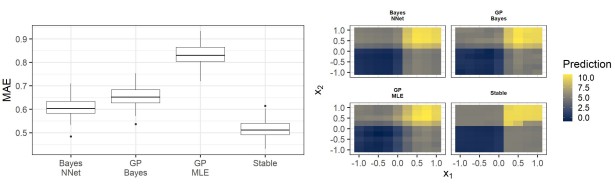

Figure 3: *Left:* Boxplots of mean absolute error (MAE) of out-of-sample prediction over test points, and *Right:* predicted values over a $9 \times 9$ grid on $[-1, 1]^2$.

We present quantiles of the posterior predictive distribution for GP Bayes and Stable methods in Figure 4. Our results show sharper jumps using the Stable method, when the true function has jump discontinuities.

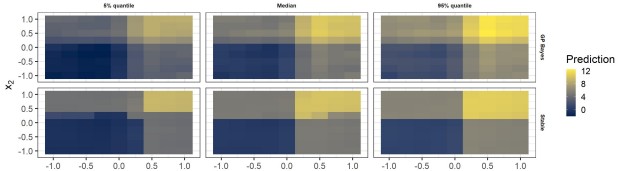

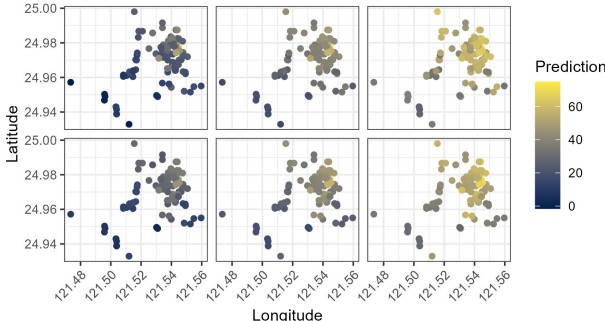

Figure 4: Posterior predictive quantiles at the $5\%$, $50\%$, and $95\%$ levels for GP Bayes (*upper*) and Stable (*lower*) over a $9 \times 9$ grid on $[-1, 1]^2$.

Figure 5: Posterior predictive quantiles at the $5\%$, $50\%$, and $95\%$ levels for GP Bayes (*upper*) and Stable (*lower*) on validation.

# 5 OUT OF SAMPLE PREDICTION ON REAL ESTATE VALUATION DATA IN TAIPEI

Valuation of real estate properties in Taipei, Taiwan were collected by Yeh and Hsu (2018) in different locations. The data are available from the UCI repository[1], and is a benchmark dataset. We apply our method to the spatial locations of the properties, to predict the valuations of the real estate dataset. We use 276 locations for training and 138 for testing. We compare the performance of our method to the three methods mentioned in the previous section through the mean absolute error. The results are displayed in Table 1, showing a competitive MAE under the proposed approach. Figure 5 displays the posterior predictive quantiles on the validation set, with narrower intervals under the proposed stable method in most cases.

Table 1: Mean absolute error of predictions by method and standard errors computed on 10 random training–testing splits in the real estate data set.

|      | Stable | GP MLE | GP Bayes | Bayes NNet |
|------|--------|--------|----------|------------|
| MAE  | 0.415  | 0.483  | 0.402    | 0.501      |
| (SE) | (0.07) | (0.07) | (0.05)   | (0.07)     |

Supplementary Sections S.5 and S.6 present results on ablation experiments and additional data sets with larger sample sizes, including recent data on S&P stock index.

# 6 CONCLUSIONS

We develop a novel method for posterior inference and prediction for infinite width limits of shallow (one hidden layer) BNNs under weights with infinite prior variance. While the $\alpha$-stable forward scaling limit in this case has been known in the literature (Der and Lee, 2005; Peluchetti et al., 2020), the lack of a covariance function precludes the inverse problem of feasible posterior inference and prediction, which we overcome using a conditionally Gaussian representation.

---

[1]https://archive.ics.uci.edu/dataset/477/
real+estate+valuation+data+set

There is a wealth of literature on the universal approximation property of both shallow and deep neural networks, following the pioneering work of Hornik et al. (1989), but they work under the assumption of a *continuous* true function. Our numerical results demonstrate that when the truth has jump discontinuities, it is possible to obtain much better results with a BNN using weights with unbounded prior variance. The fully Bayesian posterior also allows straightforward probabilistic uncertainty quantification for the infinite width scaling limit under $\alpha$-stable priors on network weights.

Several future directions could naturally follow from the current work. The most immediate is perhaps an extension to posterior inference for deep networks under stable priors, where the width of each layer simultaneously approaches infinity, and we strongly suspect this should be possible. The role of the non-degenerate posterior of $\mathbf{Q_s} \mid \mathbf{y}$ on deep generalizations and representation learning merits a thorough investigation and suggests crucial differences from a GP limit (Aitchison et al., 2021; Yang et al., 2023). Developing analogous results under non-i.i.d. or tied weights to perform posterior inference under the scaling limits for Bayesian convolutional neural networks should also be of interest. Finally, one may of course investigate alternative activation functions, such as the hyperbolic tangent, which will lead to a different characteristic function for the scaling limit.

# 7 PROOF OF THEOREM 1

Our derivations rely on Equation 5.4.6 of Uchaikin and Zolotarev (1999), which states that for $\alpha_0 \in (0, 1)$ and for all positive $\lambda$ one has: $\exp(-\lambda^{\alpha_0}) = \int_0^\infty \exp(-\lambda t) p_{S+}(t) dt$, where $p_{S+}$ is the density function of a positive $\alpha_0$-stable random variable. Using $\lambda = \nu z^2$ and $\alpha_0 = \alpha/2$ and the fact that $z^2 = |z|^2$, we obtain for $\alpha \in (0, 2)$ that:

$$\exp(-\nu^{\alpha/2}|z|^\alpha) = \int_0^\infty \exp(-\nu z^2 t) p_{S+}(t) dt, \quad (7)$$

where $p_{S^+}$ is the density function of a positive $\alpha/2$-stable random variable and $z \in \mathbb{R}$. By Equation (4) of Der and Lee (2005), and the characteristic function of independent normally distributed error terms with a common variance $\sigma^2$, we have that:

$$
\begin{aligned}
\phi_{\mathbf{y}}(\mathbf{t}) = {} & \exp\left(-\sum_{\ell=1}^{L} 2^{-\alpha/2}\nu^{\alpha/2}q_\ell \left|\sum_{j=1}^{n} t_j \tau_\ell(\mathbf{x}_j)\right|^\alpha \right. \\
& \left. -\frac{1}{2}\sum_{j=1}^{n}\sigma^2 t_j^2\right) \\
= {} & \prod_{\ell=1}^{L} \exp\left(-2^{-\alpha/2}\nu^{\alpha/2}q_\ell \left|\sum_{j=1}^{n} t_j\tau_\ell(\mathbf{x}_j)\right|^\alpha\right) \\
& \times \exp\left(-\frac{1}{2}\sum_{j=1}^{n}\sigma^2 t_j^2\right) \\
= {} & \prod_{\ell=1}^{L}\left\{\int_0^\infty \exp\left(-\frac{1}{2}\nu s_\ell q_\ell^{2/\alpha}\left(\sum_{j=1}^{n} t_j\tau_\ell(\mathbf{x}_j)\right)^2\right) \right. \\
& \left. \times p_{S^+}(s_\ell)ds_\ell\right\} \times \exp\left(-\frac{1}{2}\sum_{j=1}^{n}\sigma^2 t_j^2\right) \\
= {} & \prod_{\ell=1}^{L}\left\{\int_0^\infty \exp\left(-\frac{1}{2}\nu s_\ell q_\ell^{2/\alpha}\mathbf{t}^T\mathbf{M}_\ell\mathbf{t}\right) \right. \\
& \left. \times p_{S^+}(s_\ell)ds_\ell\right\} \times \exp\left(-\frac{1}{2}\sum_{j=1}^{n}\sigma^2 t_j^2\right),
\end{aligned}
$$

where the third equality follows by using Equation (7), and in the last equality we define $\mathbf{M}_\ell$ as the matrix with ones in the diagonal and with the $(i,j)$th entry given by $\tau_\ell(\mathbf{x}_i)\tau_\ell(\mathbf{x}_j)$, $i \neq j$. Next, using the fact that the densities are over the independent variables $\{s_\ell\}_{\ell=1}^{L}$ we bring the product inside the integrals and employ the property of the exponential to obtain:

$$
\begin{aligned}
\phi_{\mathbf{y}}(\mathbf{t}) = {} & \int_{(\mathbb{R}+)^L} \exp\left(-\frac{1}{2}\sum_{j=1}^{n}t_j^2\sigma^2 - \frac{1}{2}\nu\sum_{\ell=1}^{L}s_\ell q_\ell^{2/\alpha}\mathbf{t}^T\mathbf{M}_\ell\mathbf{t}\right) \\
& \times \prod_{\ell=1}^{L}p_{S^+}(s_\ell)ds_\ell \\
= {} & \int_{(\mathbb{R}+)^L} \exp\left(-\frac{1}{2}\mathbf{t}^T\mathbf{Q}_\mathbf{s}\mathbf{t}\right)\prod_{\ell=1}^{L}p_{S^+}(s_\ell)ds_\ell,
\end{aligned}
$$

using on the second line the definition of $\mathbf{Q}_\mathbf{s}$. The required density is now obtained by the use of the inverse Fourier transform on the characteristic function:

$$
\begin{aligned}
p(\mathbf{y}\mid\mathbf{X}) = {} & \int_{\mathbb{R}^n}\phi_{\mathbf{y}}(\mathbf{t})\exp(i\langle\mathbf{t},\mathbf{y}\rangle)\prod_{j=1}^{n}dt_j \\
= {} & \int_{\mathbb{R}^n}\left\{\int_{(\mathbb{R}+)^L}\exp\left(-\frac{1}{2}\mathbf{t}^T\mathbf{Q}_\mathbf{s}\mathbf{t}\right)\prod_{\ell=1}^{L}p_{S^+}(s_\ell)ds_\ell\right\} \\
& \times \exp(i\langle\mathbf{t},\mathbf{y}\rangle)\prod_{j=1}^{n}dt_j \\
= {} & \int_{(\mathbb{R}+)^L}\int_{\mathbb{R}^n}\exp\left(-\frac{1}{2}\mathbf{t}^T\mathbf{Q}_\mathbf{s}\mathbf{t}\right)
\end{aligned}
$$

$$
\times \exp(i\langle\mathbf{t},\mathbf{y}\rangle)\prod_{j=1}^{n}dt_j\prod_{\ell=1}^{L}p_{S^+}(s_\ell)ds_\ell,
$$

where the second line follows by the derived expression for the characteristic function, and the third line follows by Fubini's theorem since all the integrals are real and finite. We recognize that the term $\exp(-(1/2)\mathbf{t}^T\mathbf{Q}_\mathbf{s}\mathbf{t})$ corresponds to a multivariate Gaussian density with covariance matrix $\mathbf{Q}_\mathbf{s}^{-1}$, though it is lacking the usual determinant term. We obtain the density using the characteristic function of Gaussian variables to finally obtain the result:

$$
\begin{aligned}
p(\mathbf{y}\mid\mathbf{X}) = {} & (2\pi)^{-n/2}\int_{(\mathbb{R}+)^L}\exp\left(-\frac{1}{2}\mathbf{y}^T\mathbf{Q}_\mathbf{s}^{-1}\mathbf{y}\right) \\
& \times \det(\mathbf{Q}_\mathbf{s})^{-1/2}\prod_{\ell=1}^{L}p_{S^+}(s_\ell)ds_\ell.
\end{aligned}
$$

In using $\mathbf{Q}_\mathbf{s}^{-1}$ freely, we assumed through the previous steps that $\mathbf{Q}_\mathbf{s}$ is positive-definite. We proceed to prove this fact. Note that $\mathbf{Q}_\mathbf{s}$ is obtained from the sum of $L$ rank-one matrices and a diagonal matrix, where each of the rank-one matrices is $q_\ell^{2/\alpha}s_\ell\nu\boldsymbol{\tau}_\ell\boldsymbol{\tau}_\ell^T$. Let $\mathbf{w} \in \mathbb{R}^n\backslash\{0\}$. Then:

$$
\begin{aligned}
\mathbf{w}^T\mathbf{Q}_\mathbf{s}\mathbf{w} = {} & \mathbf{w}^T\left(\sigma^2\mathbf{I} + \nu\sum_{\ell=1}^{L}s_\ell q_\ell^{2/\alpha}\boldsymbol{\tau}_\ell\boldsymbol{\tau}_\ell^T\right)\mathbf{w} \\
= {} & \sigma^2\mathbf{w}^T\mathbf{w} + \nu\sum_{\ell=1}^{L}s_\ell q_\ell^{2/\alpha}\mathbf{w}^T\boldsymbol{\tau}_\ell\boldsymbol{\tau}_\ell^T\mathbf{w} \\
= {} & \sigma^2\sum_{j=1}^{n}w_j^2 + \nu\sum_{\ell=1}^{L}s_\ell q_\ell^{2/\alpha}\left(\sum_{j=1}^{n}w_j\tau_\ell(x_j)\right)^2 \\
> {} & 0,
\end{aligned}
$$

implying that $\mathbf{Q}_\mathbf{s}$ is positive-definite with probability 1.

## SUPPLEMENTARY MATERIAL

The Supplementary Material contains technical details and numerical results in pdf. Computer code is freely available at: https://github.com/loriaJ/alphastableNNet.

## ACKNOWLEDGMENTS

Bhadra was supported by U.S. National Science Foundation Grant DMS-2014371.

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

# Posterior Inference on Shallow Infinitely Wide Bayesian Neural Networks under Weights with Unbounded Variance
# (Supplementary Material)

**Jorge Loría**[1,2]                                    **Anindya Bhadra**[1]

[1]Department of Statistics, Purdue University, West Lafayette, Indiana, USA
[2]Department of Computer Science, Aalto University, Finland

## S.1   SOME RELEVANT PROPERTIES OF $\alpha$-STABLE RANDOM VARIABLES

One of the most important properties of stable random variables is the closure property (Property 1.2.1, Samorodnitsky and Taqqu, 1994), which states that if $X_i \sim S(\alpha, \nu_i, \beta_i)$ independently for $i = 1, 2$, then $X_1 + X_2 \sim S(\alpha, \xi, \gamma)$, where $\gamma = (\beta_1 \nu_1^\alpha + \beta_2 \nu_2^\alpha)/(\nu_1^\alpha + \nu_2^\alpha)$, and $\xi = (\nu_1^\alpha + \nu_2^\alpha)^{1/\alpha}$. This means that the sum of two $\alpha$-stable variables is again $\alpha$-stable. This is a generalization of the well known property of the closure under convolutions of Cauchy ($\alpha = 1$) and Gaussian ($\alpha = 2$) random variables. In terms of moments, Samorodnitsky and Taqqu (1994, Property 1.2.16) indicate that for $X \sim S(\alpha, \beta, \nu)$ with $\alpha \in (0, 2)$, we have $\mathbb{E}[|X|^r] = \infty$ for $r \geq \alpha$ and $\mathbb{E}[|X|^r]$ is finite for $0 < r < \alpha$. Specifically, this implies that $\alpha$-stable random variables have infinite variance, when $\alpha < 2$.

The property of closure under convolutions is easily generalized to the sum of a sequence of i.i.d. $\alpha$-stable variables, which gives rise to a convergence in a non-Gaussian domain, for $\alpha < 2$. Formally, the *generalized* central limit theorem (Gnedenko and Kolmogorov, 1954) proves that for i.i.d. scaled random variables with infinite variance the convergence is no longer to a Gaussian random variable. Rather the convergence is to an $\alpha$-stable random variable. A statement of the theorem is below.

**Theorem S.1.** *(Generalized central limit theorem, Uchaikin and Zolotarev, 1999, p. 62) Let $X_1, \ldots, X_n$ be independent and identically distributed random variables with cumulative distribution function $F(x)$ satisfying the conditions:*

$$1 - F(x) \sim c|x|^{-\gamma}, \; x \to \infty,$$
$$F(x) \sim d|x|^{-\gamma}, \; x \to \infty,$$

*with $\gamma > 0$. Then there exists sequences $a_n \in \mathbb{R}$ and $b_n > 0$, such that the distribution of the centered and normalized sum:*

$$Z_n = b_n^{-1} \left( \sum_{i=1}^n X_n - a_n \right),$$

*weakly converges to $S(\alpha, 1, \beta)$ as $n \to \infty$, where $\alpha = \min(\gamma, 2)$, $\beta = (c - d)/(c + d)$, and $a_n$ and $b_n$ are as given in Table S1.*

Finally, the Laplace transforms of positive $\alpha$-stable random variables exist. For $\alpha < 1$ and $X \sim S(\alpha, 1, 1)$ the Laplace transform is given by:
$$\mathbb{E}[\exp(-\lambda X)] = \exp(-\lambda^\alpha),$$

for $\lambda > 0$.

## S.2   COMPUTATION OF $q_\ell$ AND $\tau_\ell$

Since the size of $\Lambda$ is $2^n$, indicating an exponential complexity of naïve enumeration, we further simplify Equation (4). When the input points are arranged in a way that $\tau_\ell$ is not possible, then $q_\ell$ must be zero. We can identify the elements in $\Lambda$

| $\gamma$ | $a_n$ | $b_n$ |
|---|---|---|
| $\gamma \in (0,1)$ | $0$ | $[\pi(c+d)]^{1/\gamma}[2\Gamma(\gamma)\sin(\gamma\pi/2)]^{-1/\gamma}n^{1/\gamma}$ |
| $\gamma = 1$ | $\beta(c+d)n\ln(n)$ | $(\pi/2)(c+d)n$ |
| $\gamma \in (1,2)$ | $n\mathbb{E}[X]$ | $[\pi(c+d)]^{1/\gamma}[2\Gamma(\gamma)\sin(\gamma\pi/2)]^{-1/\gamma}n^{1/\gamma}$ |
| $\gamma = 2$ | $n\mathbb{E}[X]$ | $(c+d)^{1/2}[n\ln(n)]^{1/2}$ |
| $\gamma > 2$ | $n\mathbb{E}[X]$ | $[(1/2)\mathrm{Var}(X)]^{1/2}n^{1/2}$ |

with positive probabilities by considering arbitrary values of $b_0, w_1, \ldots, w_I$. The corresponding $\tau$ function is determined by: $\tau(\mathbf{x}_j) = \mathrm{sign}(b_0 + w_1 x_{j1} + \cdots + w_I x_{jI})$, for each $j$. This corresponds to labeling with $+1$ the points above a hyperplane, and with $-1$ the points that lie below the hyperplane. When $I = 1$, without loss of generality, let $x_1 < \cdots < x_n$. In this case $\Lambda$ corresponds to the possible changes in sign that can occur between the input variables, which is equal to $n$. Specifically, the sign change can occur before $x_1$, between $x_1$ and $x_2$, ..., between $x_{n-1}$ and $x_n$, and after $x_n$. A similar argument is made in Example 2.1.1 of Der and Lee (2005), suggesting the possibility of considering more than one dimensions.

For $I > 1$, Harding (1967) studies the possible partitions of $n$ points in $\mathbb{R}^I$ by an $(I-1)$-dimensional hyperplane—which corresponds to our problem, and determines that for points in general configuration there are $O(n^I)$ partitions. Goodman and Pollack (1983) give an explicit algorithm for finding the elements of $\Lambda$ that have non-zero probabilities in any possible configuration. We summarize their algorithm for $I = 2$ as Algorithm S.1 in Supplementary Section S.3. This algorithm runs in a computational time of order $n^I \log(n)$, which is reasonable for moderate $I$. For the rest of the article, we denote the cardinality of elements in $\Lambda$ that have positive probability by $L$, with the understanding that $L$ will depend on the input vectors $\mathbf{x}_i$ that are used and their dimension. This solves the issue of computing deterministically the values of $\tau_\ell$ that have positive probability after integrating through input-to-hidden weights.

Next we compute the probability $q_\ell$ for the determined $\tau_\ell$. For $I = 1$, the $q_\ell$s correspond to probabilities obtained from a Cauchy cumulative density function, which we state explicitly in Supplementary Section S.2.1. For general dimension of the input $I > 1$, the value of $q_\ell$ is given by $\mathbb{P}(\mathbf{Z}^{(\tau_\ell)} > 0)$, where the $n$-dimensional Gaussian vector $\mathbf{Z}^{(\tau_\ell)}$ has $i$-th entry given by $\tau_\ell(\mathbf{x}_i)(b_0 + \sum_{j=1}^I w_j x_{ij})$. This implies that $\mathbf{Z}^{(\tau_\ell)} \sim \mathcal{N}_n(0, \mathbf{\Sigma}^{(\tau_\ell)})$, where the (possibly singular) variance matrix is $\Sigma_{i,j}^{(\tau_\ell)} = \tau_\ell(\mathbf{x}_i)\tau_\ell(\mathbf{x}_j)(1 + \sum_{k=1}^n x_{ik}x_{jk})$. This means we can compute $q_\ell = \mathbb{P}(\mathbf{Z}^{(\tau_\ell)} > 0)$ using for example the R package `mvtnorm`, which implements the method of Genz and Bretz (2002) for evaluating multivariate Gaussian probabilities. Since the $q_\ell$s require independent procedures to be computed, this is easily parallelized after obtaining the partitions.

### S.2.1 COMPUTATION OF $q_\ell$ AND $\tau_\ell$ IN ONE DIMENSION

Assume, since we are in one dimension, that $x_1 < \cdots < x_n$. In the one-dimensional case $\Lambda$ consists of the different locations where the change in sign can be located. This corresponds to:

1. Before the first observation, which corresponds to $\tau(x_k) = 1$ for all $k$, we call this $\tau_0$.

2. Between $x_j$ and $x_{j+1}$ for some $j = 1, \ldots, n-1$, which corresponds to $\tau(x_k) = -1$ for $k < j$ and $\tau(x_k) = +1$ otherwise, we call this $\tau_j$.

3. After $x_n$, $\tau(x_k) = -1$ for all $k$, and we call this $\tau_n$.

Note that the first and last items correspond to linearly dependent vectors as $\tau_0(x_k) = -\tau_n(x_k)$, for all $k = 1, \ldots, n$. Now, we compute the probability for the first and last items:

$$
\begin{aligned}
q_{\tau_0} + q_{\tau_n} &= \mathbb{P}(b_0 + w_1 x_1 < 0) + \mathbb{P}(b_0 + w_1 x_n > 0) \\
&= \mathbb{P}(x_1 < -b_0/w_1) + \mathbb{P}(x_n > -b_0/w_1) \\
&= \mathbb{P}(x_1 < -C) + \mathbb{P}(x_n > -C) \\
&= \frac{1}{2} + \frac{1}{\pi}\arctan(x_1) + \frac{1}{2} - \frac{1}{\pi}\arctan(-x_n) \\
&= 1 + \frac{1}{\pi}(\arctan(x_1) - \arctan(-x_n)),
\end{aligned}
$$

where $C \sim \text{Cauchy}(0,1)$ since $b_0, w_1$ are independent standard normal variables.

Next, the change in sign occurring between $x_j$ and $x_{j+1}$, yields:

$$
\begin{aligned}
q_{\tau_j} &= \mathbb{P}(\text{sign}(b_0 + w_1 x_j) = -1, \ \text{sign}(b_0 + w_1 x_{j+1}) = 1) \\
&= \mathbb{P}(b_0 + w_1 x_j < 0, \ b_0 + w_1 x_{j+1} > 0) \\
&= \mathbb{P}(x_j < -b_0/w_1, \ x_{j+1} > -b_0/w_1) \\
&= \mathbb{P}(x_j < C < x_{j+1}) \\
&= \frac{\arctan(x_{j+1}) - \arctan(x_j)}{\pi},
\end{aligned}
$$

which corresponds to the desired probability.

## S.3 SUPPORTING ALGORITHMS

Algorithm S.1 is modified from Goodman and Pollack (1983) for $I = 2$ dimensions, which we employ in Algorithm 1, and has a computational complexity of $\mathcal{O}(n^I \log(n))$ for $n$ input points and a general input dimension $I$. We present Algorithm S.2 to sample the latent scales $\{s_\ell\}_{\ell=1}^L$ and error standard deviation $\sigma$, which consists of a independent samples Metropolis–Hastings procedure where we sample from the priors, and iteratively update the matrix $\mathbf{Q_s}$. For computation of the density functions we use the Woodbury formula, and an application of the matrix-determinant lemma, for an efficient update of $\mathbf{Q_s}$ and to avoid computationally intensive matrix inversions. Algorithm S.2 has computational complexity of $\mathcal{O}(Ln^2) = \mathcal{O}((n+m)^I n^2)$.

---

**Algorithm S.2** A Metropolis–Hastings sampler for $\mathbf{Q}$ by simulating the latent scales from the prior

---

**Require:** Vector $\mathbf{y} \in \mathbb{R}^n$, previous latent scales $\{s_\ell\}_{\ell=1}^L$, vectors $\{\tau_\ell\}_{\ell=1}^L$, partition probabilities $\{q_\ell\}_{\ell=1}^L$, previous variance matrix $\mathbf{Q} = \nu \sum_{\ell=1}^L s_\ell q_\ell^{2/\alpha} \tau_\ell \tau_\ell^T + \sigma^2 \mathbf{I}$, and magnitude of errors $\sigma^2$.
**Output:** Updated $\mathbf{Q}$ matrix.

    **for** $k = 1, \ldots, L$ **do**
        Propose $s_k^* \sim S^+(\alpha/2)$.
        Define $\mathbf{Q}^{(prop)} = \nu \sum_{\ell \neq k} s_\ell q_\ell^{2/\alpha} \tau_\ell \tau_\ell^T + \nu s_k^* q_k^{2/\alpha} \tau_k \tau_k^T + \sigma^2 \mathbf{I}$ .
        Accept $s_k^*$ with probability $\min\{p(\mathbf{y} \mid \mathbf{Q}_{1:n,1:n}^{(prop)})/p(\mathbf{y} \mid \mathbf{Q}_{1:n,1:n}), 1\}$.
        If $s_k^*$ is accepted, replace $s_k$ by $s_k^*$.
    **end for**
    Propose $\sigma_*^2 \sim \text{Cauchy}^+(0,1)$.
    Compute $\mathbf{Q}^{(prop)} = \nu \sum_{\ell=1}^L q_\ell^{2/\alpha} s_\ell \tau_\ell \tau_\ell^T + \sigma_*^2 \mathbf{I}$.
    Accept $\mathbf{Q}^{(prop)}$ with probability $\min\{p(\mathbf{y} \mid \mathbf{Q}_{1:n,1:n}^{(prop)})/p(\mathbf{y} \mid \mathbf{Q}_{1:n,1:n}), 1\}$.
    **return** $\mathbf{Q}^{(prop)}$ if it was accepted, **return** $\mathbf{Q}$ otherwise.

---

## S.4 ADDITIONAL NUMERICAL RESULTS

We report MCMC convergence diagnostics, run times, and posterior quantiles of $\mathbf{Q_s} \mid \mathbf{y}$. We also include simulation results for a variety of functions in one and two dimensions, where we demonstrate the flexibility of the proposed method.

### S.4.1 MCMC DIAGNOSTICS AND COMPUTATION TIMES

### S.4.2 POSTERIOR QUANTILES OF $\mathbf{Q_s} \mid \mathbf{y}$

We present results showing the posterior $\mathbf{Q_s} \mid \mathbf{y}$ is non-degenerate under a stable limit. In the case of a vanilla GP limit the prior on the kernel converges to a point mass, resulting in a degenerate posterior (Aitchison et al., 2021; Yang et al., 2023). However, as shown by our Corollary 1, $\mathbf{Q_s}$ is stochastic under a stable limit. Figure S2 displays the posterior 25th, 50th, and 75th quantiles of $\mathbf{Q_s} \mid \mathbf{y}$ for the examples shown in Sections 4.1 and 4.2, confirming the posterior of $\mathbf{Q_s} \mid \mathbf{y}$ is non-degenerate. This is a key feature that distinguishes the current work from prior works on GP limits.

**Algorithm S.1** (Goodman and Pollack, 1983). Multidimensional sorting for $I = 2$

**Require:** Matrix $\mathbf{X} \in \mathbb{R}^{n \times 2}$.
**Output:** partition vectors $\{\tau_\ell\}_{\ell=1}^L$.

    **for** $i = 1, \ldots, n-1$ **do**

        **for** $j = i+1, \ldots, n$ **do**

            Let $u_j = x_{j,1} - x_{i,1}$, and $v_j = x_{j,2} - x_{i,2}$. If $(u_j, v_j) = (0,0)$ call $j$ "good".

            Let $u_{n+j} = -u_j$, $v_{n+j} = -v_j$, and let $m_j = m_{n+j} = v_j/u_j$.

        **end for**

        Sort the indices $\{j : j \text{ is good}\} \cup \{n+j : j \text{ is good}\}$ into subsets:

        1. for those for which $u_j > 0$, using $m_j$ as key

        2. for those for which $u_j = 0$ and $v_j > 0$

        3. for those for which $u_j < 0$, using $m_j$ as key

        4. for those for which $u_j = 0$, and $v_j < 0$

        From the sorting in Items 1 and 3 we obtain a list of subsets. Say: $\{J_{11}, \ldots, J_{1p_1}, \ldots, J_{r1}, \ldots, J_{rp_r}\}$, where the points with indices $J_{k1}, \ldots, J_{kp_k}$ constitute an entire subset, and denote $J^{(k)}$ as their union, and there are $r$ subsets all together. Denote by $k(j)$ the number of the subset within which $j$ lies.

        For each $k = 1, \ldots, r$, let: $n_k = \#\{m : 1 \leq m, J_{km} \leq n\}$, the number of points in each ray.

        For each good $j$, consider $A_0^{(ij)} = \{i\} \cup (J^{(k(j))} - \{n+1, \ldots, 2n\})$ as the points in the same ray as $ij$.

        **if** $k(n+j) > k(j)$ **then**

            Define the points in the positive side by: $A_+^{(ij)} = \cup_{k=k(j)+1}^{k(n+j)-1} J^{(k)} - \{n+1, \ldots, 2n\}$, the points in the negative

    side by: $A_-^{(ij)} = \{1, \ldots, n\} - A_+^{(ij)} - A_0^{(ij)}$.

        **else if** $k(n+j) < k(j)$ **then**

            Define the points in the positive side by: $A_+^{(ij)} = \cup_{k=k(j)+1}^{r} J^{(k)} \cup \cup_{k=1}^{k(n+j)-1} J^{(k)}$, and the points in the negative

    side by: $A_-^{(ij)} = \{1, \ldots, n\} - A_+^{(ij)} - A_0^{(ij)}$.

        **end if**

        For each $j = i+1, \ldots, n$, if $A_0^{(ij)}$ has $L_j$ ordered items denoted by $\{a_m : m = 1, \ldots, L_j\}$. Add $2L_j$ vectors: $\tau_\ell$ for $\ell = 1, \ldots, L_j$, with entry $k$ equal to $+1$ if $k \in A_+^{(ij)} \cup \{a_m : m = 1, \ldots, \ell\}$, and $-1$ otherwise, and the vectors $\tau_{\ell+L_j}$ for $\ell = 1, \ldots, L_j$, with entry $k$ equal to $+1$ if $k \in A_+^{(ij)} \cup \{a_m : m = \ell+1, \ldots, L_j\}$, and $-1$ otherwise.

        Repeat using $A_+^{(ij)}$ as the negative, and $A_-^{(ij)}$ as the positive.

    **end for**

    Discard repeated vectors.

    **return** the collection of vectors $\{\tau_\ell\}_{\ell=1}^L$

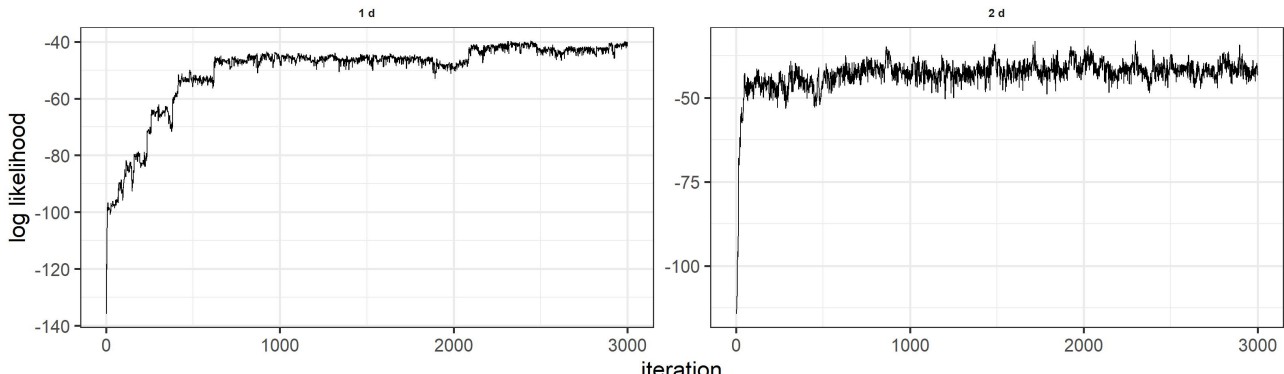

Figure S1: Trace plots for the proposed MCMC sampler (Algorithm 1) for the simulations in Section 4, indicating good mixing in about 1000 burn-in iterations. *Left:* one dimensional case, *Right:* two dimensional case. Numerical results were obtained using the last 2000 iterations.

Table S2: Total (in seconds) and per iteration (in milliseconds) Computation Times for the Simulations in Section 4, for the Competing Methods.

|  |  | Stable | GP MLE | GP Bayes | Bayes NNet |
|---|---|---|---|---|---|
| Total time (s) | 1-d | 24.047 | 0.067 | 18.073 | 6.91 |
|  | 2-d | 1339.543 | 0.178 | 22.185 | 7.193 |
| Per iteration time (ms) | 1-d | 8.016 | 13.400 | 0.602 | 2.303 |
|  | 2-d | 446.514 | 35.600 | 0.740 | 2.398 |

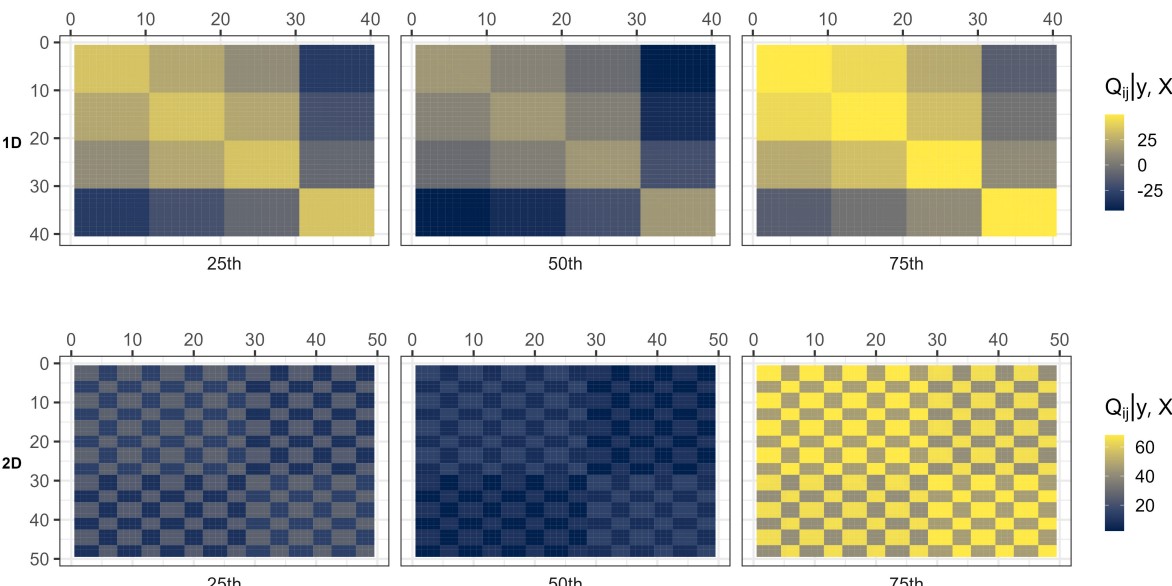

Figure S2: Posterior quantiles (*left:* 25, *center:* 50, *right:* 75) of the kernel $\mathbf{Q_s}$, for the 1-d (*upper*) and 2-d (*lower*) examples, clearly showing a non-degenerate posterior for $\mathbf{Q_s}$.

### S.4.3   ADDITIONAL RESULTS IN ONE DIMENSION

We show, using a variety of one-dimensional functions, that the Stable procedure results in better performance in presence of discontinuities, while performing similarly to GP-based methods or finite width networks for smooth functions. Posterior uncertainty quantification results are omitted.

**One-dimensional one-jump function.**   Consider the function with a single jump given by $f(x) = 5 \times \mathbf{1}_{\{x>0\}}$. We use forty equally-spaced observations between $-2$ and $2$ with a Gaussian noise with standard deviation of $0.5$. We display the obtained results on Figure S3, with optimal hyper-parameters $\alpha^* = 1.1$ and $\nu^* = 1$.

**One-dimensional two-jump function.**   Consider the function with two jumps given by $f(x) = 5 \times \mathbf{1}_{\{-2/3 \leq x < 2/3\}}$. We use forty equally-spaced observations between $-2$ and $2$ with a Gaussian noise with standard deviation of $0.5$. We display the obtained results on Figure S4, with optimal hyper-parameters $\alpha^* = 1.3$ and $\nu^* = 1$.

**One-dimensional piece-wise smooth.**   Consider the piece-wise smooth function with a single jump, given by

$$f(x) = \begin{cases} -2x^2 + 8, & x \geq 0, \\ -3x + 2, & x < 0. \end{cases}$$

We use forty equally-spaced observations between $-2$ and $2$ with a Gaussian noise with standard deviation of $0.5$, and display the obtained results in Figure S5, using the optimal hyper-parameters $\alpha^* = 1$ and $\nu^* = 1$.

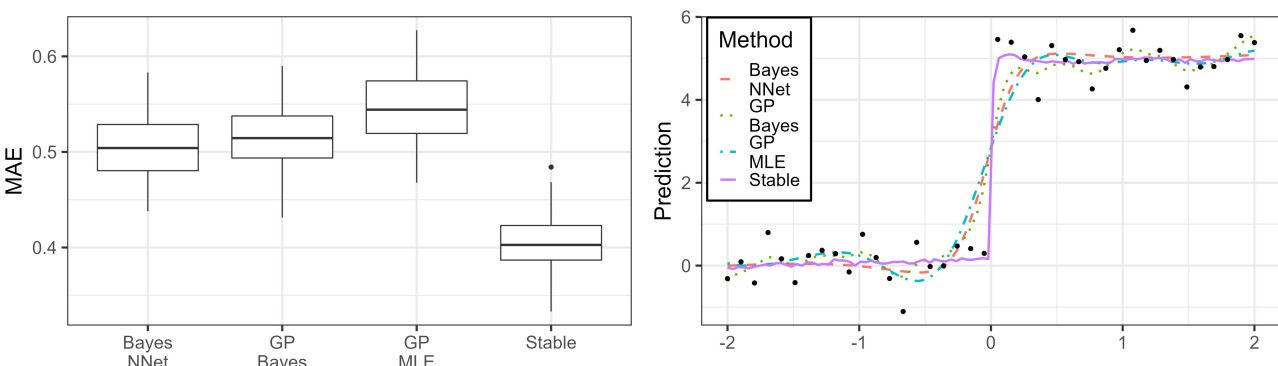

Figure S3: Out-of-sample error comparison and predictions with scatter plot of the observations for the four methods for a function with a single jump.

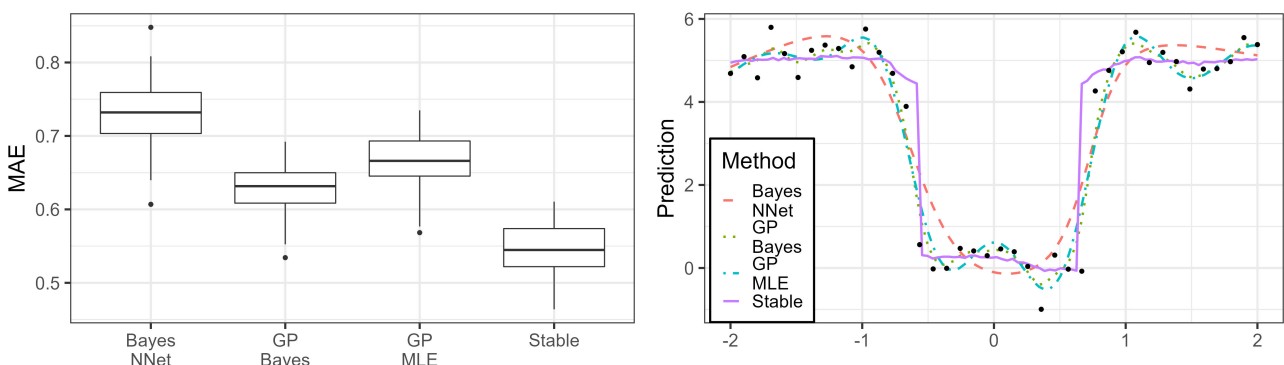

Figure S4: Out-of-sample error comparison and predictions with scatter plot of the observations for the four methods for a function with two jumps.

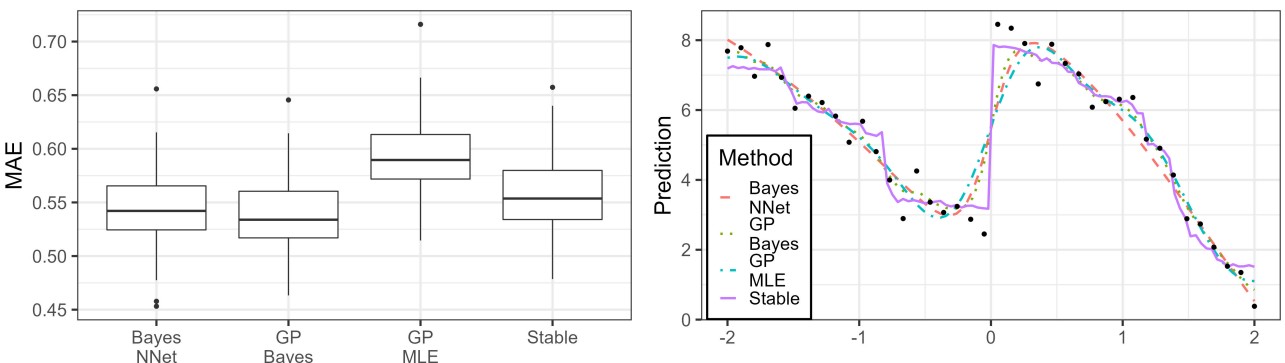

Figure S5: Out-of-sample error comparison and predictions with scatter plot of the observations for the four methods for a piece-wise smooth function.

**One-dimensional smooth function.** Finally, consider the smooth function $f(x) = -2\cos(x)^2 + 3\tanh(x) - 2x$. We use forty equally-spaced observations between $-2$ and $2$ with a Gaussian noise with standard deviation of $0.5$. The obtained results are shown in Figure S6. The optimal hyper-parameters are $\alpha^* = 1.9$ and $\nu^* = 1$.

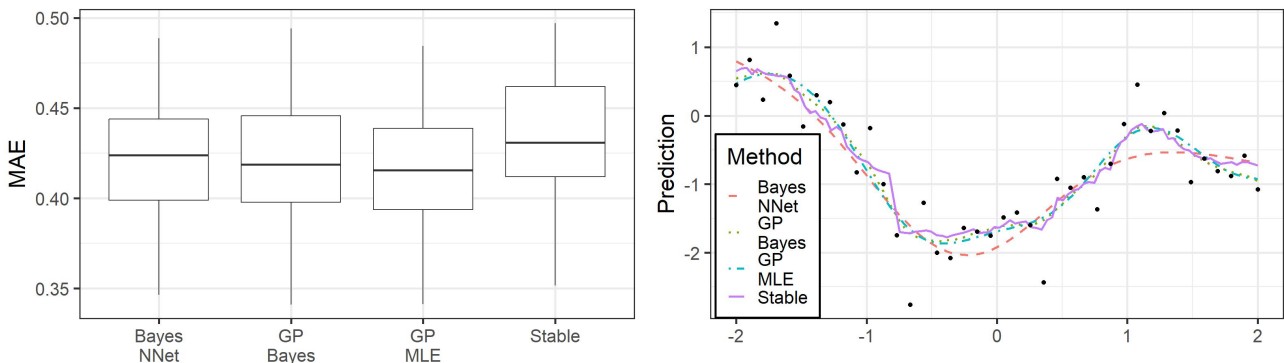

Figure S6: Out-of-sample error comparison and predictions with scatter plot of the observations for the four methods for a smooth function.

### S.4.4  ADDITIONAL RESULTS IN TWO DIMENSIONS

We show, using a variety of two-dimensional functions, that the Stable procedure results in better performance in presence of discontinuities, while performing similarly to GP-based methods or finite width networks for smooth functions. Posterior uncertainty quantification results are available, but omitted.

**Two-dimensional one-jump function.**   Consider the function $f(x_1, x_2) = 5 \times \mathbf{1}_{\{x_1 + x_2 > 0\}}$. Using the grid of points on $[-1, 1]^2$ detailed in Section 4, and additive Gaussian noise with $\sigma = 0.5$, we obtain the predictions results as shown in Figure S7. Optimal hyper-parameters are $\alpha^* = 0.1$ and $\nu^* = 1$.

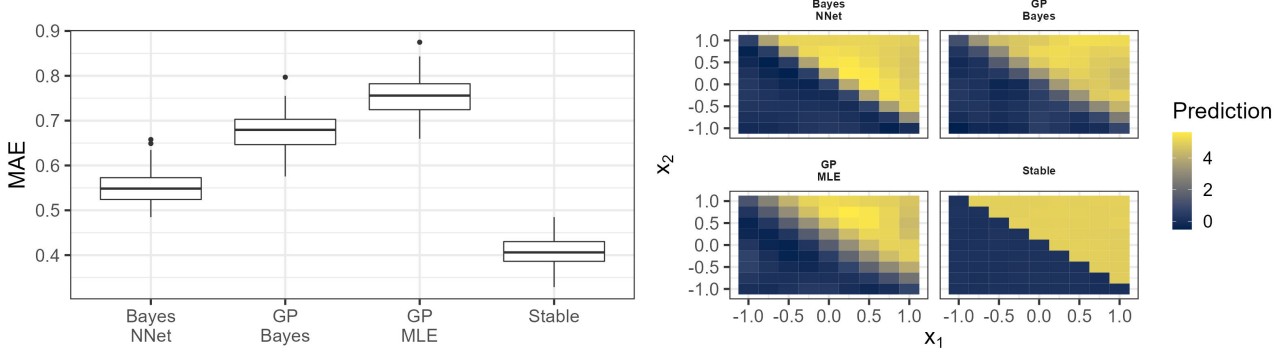

Figure S7: Out-of-sample error comparison and predictions for the four methods for a jump function in two-dimensions.

**Two-dimensional smooth edge.**   Consider the function $f(x_1, x_2) = 5 \times \mathbf{1}_{\{x_1^2 + 2x_2 - 0.4 > 0\}}$. Note that the jump boundary is determined by a smooth curve. Using the grid of points on $[-1, 1]^2$ detailed in Section 4, and additive Gaussian noise with $\sigma = 0.5$, we obtain the predictions results as shown in Figure S8. Optimal hyper-parameters are $\alpha^* = 1$ and $\nu^* = 1$. The Stable method is able to capture the smoothness of the jump boundary without losing predictive power.

**Two-dimensional smooth function.**   Consider the smooth function $f(x_1, x_2) = x_1^2 + x_2^2 - x_1 x_2$. We use the grid of points on $[-1, 1]^2$ detailed in Section 4, and additive Gaussian noise with $\sigma = 0.5$. Since this function is continuous, it would be expected that the Stable method would perform similarly as the competing methods. We obtain the predictions results as shown in Figure S9. The optimal hyperparameters are $\alpha^* = 1.3$ and $\nu^* = 1$.

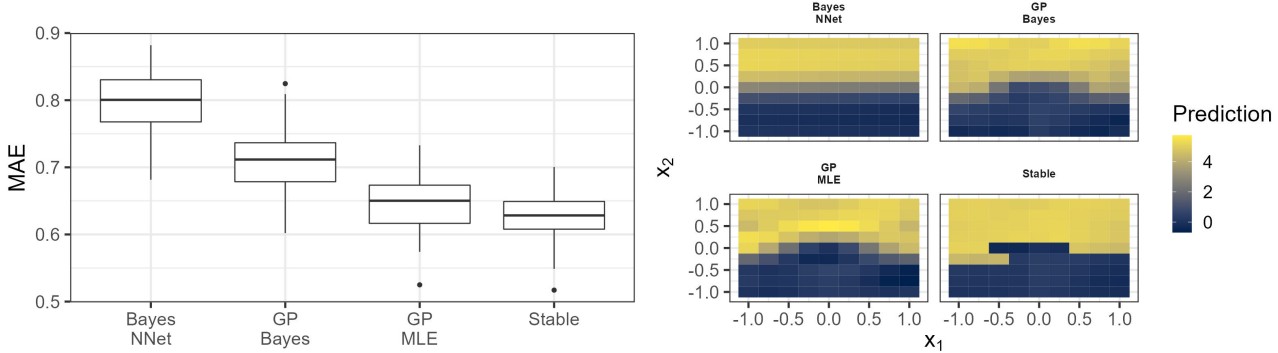

Figure S8: Out-of-sample error comparison and predictions for the four methods for a function with a jump that is parameterized by a smooth function.

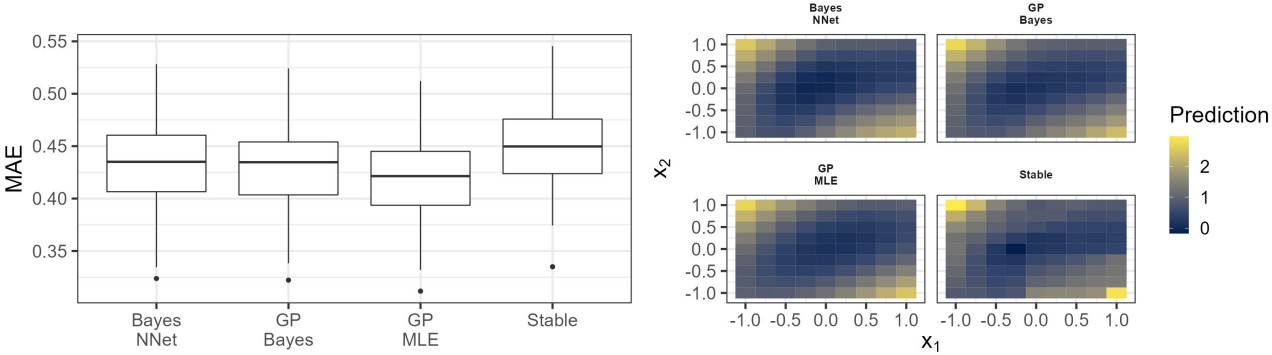

Figure S9: Out-of-sample error comparison and predictions for the four methods for a smooth function in two-dimensions.

## S.5   ABLATION STUDY ON $\alpha$ AND $\nu$

We perform an ablation study on the tuning parameters $\alpha$ and $\nu$ for the examples we consider in Section 4. Figure S10 displays the mean absolute error for a grid of the tuning parameters in all two examples. The results show that the smaller $MAE$s are mostly concentrated close to $\nu = 1$. These results hint that $\nu = 1, \alpha = 1$ are good default choices.

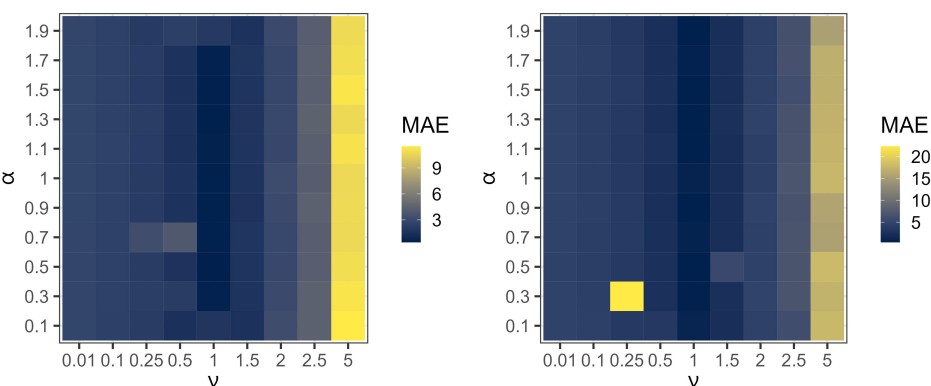

Figure S10: Ablation study for the two numerical examples. Displaying the mean absolute error for varying $\alpha$ and $\nu$ parameters. *Left:* one dimension. *Right:* two dimensions.

## S.6   RESULTS ON S&P 500

We provide out of sample prediction results for S&P 500 closing prices[1], using 336 and 169 as the training and testing set sizes respectively, with $\alpha^* = 1.9$ and $\nu^* = 1$. Table S3 displays the improved performance from using the Stable method compared to the three competing methods, as measured by the mean absolute error.

Table S3: Mean absolute error of predictions and standard errors computed on 10 random training–testing splits in the S&P 500 closing prices by method.

|  | Stable | GP MLE | GP Bayes | Bayes NNet |
|---|---|---|---|---|
| MAE | 0.054 | 0.071 | 0.054 | 0.210 |
| (SE) | (0.008) | (0.006) | (0.005) | (0.016) |

---

[1]Obtained from `https://www.nasdaq.com/market-activity/index/spx/historical` between July 1, 2019 and June 30, 2021.