# OpenReview forum: "Posterior Inference on Shallow Infinitely Wide Bayesian Neural Networks under Weights with Unbounded Variance"
_auai.org/UAI/2024/Conference — UAI 2024 poster_

### Official Review · Reviewer_CVVQ · 2024-03-20

**Q2-1 Originality-Novelty:** 2
**Q2-2 Correctness-Technical Quality:** 3
**Q2-5 Clarity Of Writing:** 3

**Q1 Summary And Contributions:**

This paper gives a complete method to sample from the posterior of Bayesian Neural Network, with one hidden layer, using a generalization of the infinite width limit approximation by Gaussian process. The proposed method works for alpha-stable weights. Prior works (like [Der and Lee, 2005]) gave limit processes in that case. The authors obtain an explicit formula for the pdf of $y|x$ when $y=f(x) +$ noise, and combine this results with a Metropolis-Hastings sampler to sample from the posterior distribution.
The results are illustrated with numerical experiments on simulated and real datasets.

**Q2-3 Extent To Which Claims Are Supported By Evidence:**

4: Excellent: all claims are supported by very convincing evidence (in the form of comprehensive experimental evaluation, rigorous mathematical proofs, detailed (pseudo-)code, precise references, well-motivated and realistic assumptions) and the authors deliver what they promise.

**Q2-4 Reproducibility:**

4: Excellent: key resources (e.g. proofs, code, data) are available and key details (e.g. proof sketches, experimental setup) are comprehensively described for competent researchers to confidently and easily reproduce the main results.

**Q3 Main Strengths:**

The paper is well written, with nice introduction giving references to the existing literature.
The mathematical result (Th. 1) seems to be rigorously derived.
The authors give different research directions in the conclusion.

**Q4 Main Weakness:**

This method is computationally prohibitive in high dimension (see the discussion in Sec 2.1).

**Q5 Detailed Comments To The Authors:**

In Prop 1, no assumption on g is written. The assumption of Der and Lee should be written, such that the reader do not have to check in the paper of Der and Lee. What do you mean by « proper distribution » ? It should be defined.
Similar remark for Th. 1 : please give clearly the precise assumptions.

It is mentioned in the abstract that the proposed procedure is « computationally efficient ». Could you give details about this statement ?

**Q9 Complying With Reviewing Instructions:**

Yes

---

> ### Author Rebuttal · Authors · 2024-04-04
>
> We thank the referee for constructive comments, for asking us to clarify ceetain details. We find most of your concerns addressable and hope you will consider updating your score.
>
> **Q4**
>
> Your point regarding computational cost is well noted. However, we note the following, in connection to usual GP regression:
>
> - Even for GP computation, there is a $O(n^3)$
>  term, so in many problems of practical interest, such as spatial (2-dimensional) or spatiotemporal (3-dimensional) data sets, we have similar complexity as GP.
>
> - GPs could also have problems in large input dimensions, unless the covariance function is simplified to have special structures (e.g., a Kronecker product along the input dimensions). Further, designing a valid positive semi-definite cross-covariance function in a GP is non-trivial when the input dimension is large.
>
> - Regardless of the ease or difficulty of computation, there are clear situations where a GP limit is inadequate. Two of these are (a) when the true function is discontinuous, on which a GP prior puts zero probability and (b) when representation learning is desired, which is impossible under a GP. Hence, there are clear motivations for considering non-GP limits.
>
>  - We suspect that the complexity may be reduced if we use activation functions other than the sign function. The main reason we used the sign function is (a) it is easy to compare with existing results of Der and Lee (2005) and Neal (1996), and (b) the separating hyperplanes resulting from this activation are intuitive to understand. But convergence to a stable process occurs with other bounded activations, such as tanh, and it is not a crucial aspect of our work. The application of the stable CLT by Gnedenko and Kolmogorov only requires that the activation function is bounded. Thus ReLU might require more work, since it is not bounded. However, a detailed study of this conjecture requires computing the analogous
>  term with other activations, and is beyond the scope of the current paper.
>
>  **Q5**
>
>  1. We will add a section in the supplement providing the detailed assumptions of Der and Lee (2005). This will not be hard to do in the camera-ready.
>
>  2. *proper distribution on input-to-hidden weights* here should be changed to just *random input-to-hidden weights*, as we need randomness in the weight-space for a *Bayesian* neural network. We apologize and this is what we wanted to convey.
>
>  3. Please see response to Q5.1 above and we will clarify all assumptions needed for Theorem 1 in the same supplementary section.
>
>  4. The procedure is the first that is *computationally feasible* for performing posterior inference in this setting and accordingly, we will change *computationally efficient* to *computationally feasible*. We accept that whether or not it is *efficient* can be debated. Nevertheless, we believe our paper is that critical first step on which future works can build.
>
>  **Q7**
>
> We also note that our works are not merely extensions of Der and Lee (2005), as they did not consider *posterior inference.* This is the key contribution of the current work, and to our knowledge, the first work to consider this for neural networks with *even for one hidden layer*, and under *any type of activation functions.* The fact that the matrix $\mathbf{Q}_s$ is random, also hints at possibilities for learning the posterior of this kernel in a data-dependent way. We hope subsequent works can build on the platform we have laid.

---

### Official Review · Reviewer_dZ75 · 2024-03-23

**Q2-1 Originality-Novelty:** 3
**Q2-2 Correctness-Technical Quality:** 3
**Q2-5 Clarity Of Writing:** 3

**Q1 Summary And Contributions:**

The paper studies a setting where Bayesian neural network weights can have unbounded variance, leading to non-Gaussian behavior in the infinite width limit, and individual hidden nodes making large contributions. The non-Gaussian predictive density p(y|X) is modeled in terms of a Gaussian conditional density p(y|X,Q) involving additional variables Q which are then marginalized over, and proposes a MCMC algorithm which iteratively sampling Q.  The paper compares this algorithm to baseline GP and BNN methods on a range of tasks.

**Q2-3 Extent To Which Claims Are Supported By Evidence:**

3: Good: the main claims are supported by convincing evidence (in the form of adequate experimental evaluation, proofs, (pseudo-)code, references, assumptions).

**Q2-4 Reproducibility:**

4: Excellent: key resources (e.g. proofs, code, data) are available and key details (e.g. proof sketches, experimental setup) are comprehensively described for competent researchers to confidently and easily reproduce the main results.

**Q3 Main Strengths:**

The paper is well-written, albeit quite dense at some parts.

Tackling the non-Gaussian regime seems well-motivated by real-world settings where functions can change discontinously or suddenly, and the smoothness of GPs will not apply, and the empirical performance on real-world datasets corroborates this.

While I am not closely familiar with this area, the proposed algorithm is (to my knowledge) original, and the strategy of marginalizing over a conditionally Gaussian density seems an appropriate and effective way to leverage Gaussian machinery in the non-Gaussian regime.

The experimental results are relatively strong, showing improvement over other GP or BNN methods on synthetic function learning tasks and a real estate dataset (in which, as I understand it, property valuations can change sharply with location).  The supplemental material includes a range of additional experimental results

(Note: I am not an expert in this area and may be unaware of relevant baseline methods.)

**Q4 Main Weakness:**

While the proposed method does improve over the baselines, the mean absolute error improvement (in particular over GP Bayes) on the real estate (and stock price) datasets is small, and it’s not clear how much uncertainty there is in this figure given the dataset sizes.

The paper does not compare to deeper neural network models (the BNN baseline is two-layer), which could be more expressive and better model jumps / discontinuities.

It also wasn’t quite clear how the approach compared empirically to alternatives in terms of computational cost.

**Q5 Detailed Comments To The Authors:**

I found section 2 to be dense and somewhat difficult to follow – incorporating more comments on interpretability or intuition for the overall approach could be helpful.  (E.g. how to think about the meaning of Q_s, and the role it is playing?)

In section 2, it could be helpful to more explicitly alpha-stable random variables, characteristic functions, the stable limit, etc., for those less familiar with these quantities.

**Q9 Complying With Reviewing Instructions:**

Yes

---

> ### Author Rebuttal · Authors · 2024-04-04
>
> Thank you for your overall feedback and for the opportunity to provide further clarifications. We hope you can reconsider your score.
>
> **Q4**
>
> 1. We respectfully disagree that the improvements are small, as we are able to see that the differences between various estimates are still high compared to bootsrapped standard error estimates, i.e., they are *statistically significant*. These will be added to the camera ready version. Moreover, please also see the significant improvements over alternative methods in synthetic data under discontinuous functions.
>
> 2. Since the infinite-width limit of a Bayesian DNN is still a deep *Gaussian* process, it will still fail to capture jumps, as no GP can capture these, because such discontinuous functions do not belong to the reproducing kernel Hilbert space (RKHS) of a GP. Moreover, for a fair comparison, we considered both GP and non-GP stable limits under one layer case. However, we acknowledge your point that an extension to more than one hidden layers can be considered, as we also noted in the Conclusions (Section 6).
>
> 3. Your point regarding computational cost is well noted. However, we note the following, in connection to usual GP regression:
>
> - Even for GP computation, there is a $O(n^3)$
>  term, so in many problems of practical interest, such as spatial (2-dimensional)
> ) or spatiotemporal (3-dimensional) data sets, we have similar complexity as GP.
>
> - GPs could also have problems in large input dimensions, unless the covariance function is simplified to have special structures (e.g., a Kronecker product along the input dimensions). Further, designing a valid positive semi-definite cross-covariance function in a GP is non-trivial when the input dimension is large.
>
> - Regardless of the ease or difficulty of computation, there are clear situations where a GP limit is inadequate. Two of these are (a) when the true function is discontinuous, on which a GP prior puts zero probability and (b) when representation learning is desired, which is impossible under a GP. Hence, there are clear motivations for considering non-GP limits.
>
>  - We suspect that the complexity may be reduced if we use activation functions other than the sign function. The main reason we used the sign function is (a) it is easy to compare with existing results of Der and Lee (2005) and Neal (1996), and (b) the separating hyperplanes resulting from this activation are intuitive to understand. But convergence to a stable process occurs with other bounded activations, such as tanh, and it is not a crucial aspect of our work. The application of the stable CLT by Gnedenko and Kolmogorov only requires that the activation function is bounded. Thus ReLU might require more work, since it is not bounded. However, a detailed study of this conjecture requires computing the analogous
>  term with other activations, and is beyond the scope of the current paper.
>
>
> **Q5**
>
> - We will update Sec. 2.0 and 2.1 to make them clearer. The parts after Prop. 1  till end of Sec 2.1 are details that can be moved to the supplement to help the flow. The term $\mathbf{Q}_s$ is the covariance kernel in our *conditionally Gaussian* representation. An interesting feature is that this term is stochastic for $\alpha<2$ and becomes deterministic when $\alpha\to 2$. Thus, one may view the data-dependent posterior of this term akin to a feature representation learned from the data.
>
> - We will also add a section in the supplement providing additional background on $\alpha$-stable random variables. Once again, this is addressable in the camera ready.

---

### Official Review · Reviewer_z4Fb · 2024-03-23

**Q2-1 Originality-Novelty:** 3
**Q2-2 Correctness-Technical Quality:** 3
**Q2-5 Clarity Of Writing:** 3

**Q10 Ethical Concerns:**

No ethical concerns.

**Q1 Summary And Contributions:**

This work generalizes the well-known results about the scaling-limits of bayesian neural networks to unbounded weight variances. In this case, the scaling limit is an α-stable process under certain conditions. Hence studying posterior inference under the non-Gaussian α-stable scaling limit is of interest. This work claims to propose a computationally efficient MCMC procedure for posterior inference.

**Q2-3 Extent To Which Claims Are Supported By Evidence:**

2: Fair: the main claims are somewhat supported by evidence (but the experimental evaluation may be weak, or does not match entirely with the claims, important baselines may be missing, proofs contain important ideas but lack rigor, algorithmic details are only discussed superficially, references are imprecise, assumptions are not sufficiently motivated or explicated, etc.).

**Q2-4 Reproducibility:**

2: Fair: key resources (e.g. proofs, code, data) are unavailable but key details (e.g. proof sketches, experimental setup) are sufficiently well-described for an expert to confidently reproduce the main results.

**Q3 Main Strengths:**

1. This paper is well-written and well-structured.
2. The proposed method is well-motivated and effectively explained, making it easy to understand

**Q4 Main Weakness:**

1. It is not entirely clear to me what is the advantage of the considering priors with unbounded variance. More discussion on this would be beneficial. It is mentioned BNN using the proposed method obtains better results when there are jump discontinuities in the targets. But its hard to judge the validity of this claim when details of the experimental setup are missing.
2. All the tables are missing standard deviations.
3. I don't believe this method can be scaled to any realistic problems with large input dimensions.

**Q5 Detailed Comments To The Authors:**

Can you discuss further about the advantage of the considering priors with unbounded variance.

**Q9 Complying With Reviewing Instructions:**

Yes

---

> ### Author Rebuttal · Authors · 2024-04-04
>
> Thank you for your overall positive feedback. We hope you will reconsider your overall score in the light of our rebuttal. Specific responses follow.
>
> **Q4**
>
> 1. The main advantage of using non-Gaussian weights is the convergence to a *non-Gaussian* stable limit, which can capture a far richer class of functions than a GP (e.g., functions with jump discontinuities). The technical reasons are founded in probability theory, which show the function space captured under a GP to be equivalent to the Reproducing Kernel Hilbert Space (RKHS) of the covariance kernel. For example, the RKHS of the squared-exponential kernel is the space of infinitely smooth functions, while the RKHS of Mat\'ern is the Sobolev space. Nevertheless, these are all spaces of *continuous* functions, and GPs cannot depart from them, but stable processes can. Thus, when there is an abrupt change or *shock* in the data (e.g., in financial data due to market crash), GPs are not good models. We have tried to give ample demonstrations of this fact numerically using various functions. Please also see additional results in the supplement.
>
> 2. We will report standard deviations in the camera-ready. Will not be hard to make this change.
>
> 3. Your point regarding input dimension is well noted. However, we note the following, in connection to usual GP regression:
>
> - Even for GP computation, there is a $O(n^3)$
>  term, so in many problems of practical interest, such as spatial (2-dimensional) or spatiotemporal (3-dimensional) data sets, we have similar complexity as GP.
>
> - GPs could also have problems in large input dimensions, unless the covariance function is simplified to have special structures (e.g., a Kronecker product along the input dimensions). Further, designing a valid positive semi-definite cross-covariance function in a GP is non-trivial when the input dimension is large.
>
> - Regardless of the ease or difficulty of computation, there are clear situations where a GP limit is inadequate. Two of these are (a) when the true function is discontinuous, on which a GP prior puts zero probability and (b) when representation learning is desired, which is impossible under a GP. Hence, there are clear motivations for considering non-GP limits.
>
>  - We suspect that the complexity may be reduced if we use activation functions other than the sign function. The main reason we used the sign function is (a) it is easy to compare with existing results of Der and Lee (2005) and Neal (1996), and (b) the separating hyperplanes resulting from this activation are intuitive to understand. But convergence to a stable process occurs with other bounded activations, such as tanh, and it is not a crucial aspect of our work. The application of the stable CLT by Gnedenko and Kolmogorov only requires that the activation function is bounded. Thus ReLU might require more work, since it is not bounded. However, a detailed study of this conjecture requires computing the analogous
>  term with other activations, and is beyond the scope of the current paper.
>
> **Q5**
>
> Please see response to Q4.1

---

### Official Review · Reviewer_dvz7 · 2024-03-23

**Q2-1 Originality-Novelty:** 3
**Q2-2 Correctness-Technical Quality:** 3
**Q2-5 Clarity Of Writing:** 4

**Q1 Summary And Contributions:**

The main focus of the paper is to discuss posterior inference in infininte-width shallow BNNs with weights that have unbounded prior variance. The paper derives an expression for the characteristic function of the observational model, shows that it can be interpreted as a hierarchical GP with stochastic covariance matrices. Then there is an MCMC sampler for posterior described, followed by 1D and 2D experiments and UCI data.

**Q2-3 Extent To Which Claims Are Supported By Evidence:**

3: Good: the main claims are supported by convincing evidence (in the form of adequate experimental evaluation, proofs, (pseudo-)code, references, assumptions).

**Q2-4 Reproducibility:**

4: Excellent: key resources (e.g. proofs, code, data) are available and key details (e.g. proof sketches, experimental setup) are comprehensively described for competent researchers to confidently and easily reproduce the main results.

**Q3 Main Strengths:**

The paper considers the problem of posterior inference for BNNs with alpha-stable priors, this is an open problem, in addition, there are intermediate results on conditionally Gaussian representation, that can be useful in future as well, so the novelty is adequate.

Technically, the paper derives the characteristic function of the observation model with description how to make the enumeration computation more effective and parallelizable. Then, the model is generalized with added noise leading to conditionally Gaussian hierarchical model, with additionally described properties of non-degenerate posterior, interpretability of the model. Overall, these statements look rigorously derived.

I like that the authors are intepreting and explaining their theoretical results in general language. I fell that there is still some introductory material is missing around Proposition 1 which could be potentially added into appendix, but overall, this is a good presentation for such theoretical paper.

There are 1D and 2D synthetic experimental results with jump discontinuities, that show better predictions than GP and finite shallow BNN models. UCI and S&P500 experiment show lower mean absolute error of predictions. Experiments show some applicability, but there is more in weaknesses section about them.

**Q4 Main Weakness:**

I think that there are some gaps from practical usefulness.
In some of the experiments in the appendix, model looks noisy or overfitted for smooth data, with larger MAE. This is not expalined in text of the paper, and potentially some way of addressing this, theoretically, or practically, is required. There is a large gap in applicability of most GP methods for large scale models and data, based on computational times, the model is not also particularly useful for such type of the data.
Aside from MAE, it would be good to have some uncertainty metric for assessing the predicted variances.

**Q5 Detailed Comments To The Authors:**

The activation function g(·) corresponds to the sign function - please elaborate on this, how it affects the results and how this can be generalized.
Can this method be expanded to more layers straightforwardly, or what are the main problems for it?

**Q9 Complying With Reviewing Instructions:**

Yes

---

> ### Author Rebuttal · Authors · 2024-04-04
>
> Thank you for acknowledging the primacy and novelty of our work. We concur with your view that this paper lays a foundation in a new area on which future papers can build. We find most of your concerns addressable and specific responses follow.
>
> **Q4**
>
> You are correct that the fit is somewhat more noisy compared to a GP *for smooth functions.* The reason is, for smooth functions, a GP is an adequate model. But a GP is inappropriate if the true function contains discontinuities, as we have demonstrated via extensive simulations. The more technical reason for this is that the functions a GP can capture is determined by the Reproducing Kernel Hilbert Space (RKHS) of the covariance kernel, which, being a Hilbert space, does not allow jump discontinuities. Thus, the method is far superior in cases GPs do not work, while *not being much worse* even for smooth functions. We view this as a strength.
>
> Similarly, although computational times are certainly a concern, the statistical performance always takes precedence, and here we comfortably outperform GP or BNN is most settings.
>
> We did provide full uncertainly quantification via posterior credible intervals; see for example Figures 2 and 5. We will also report summary such as posterior mean and variance in the camera-ready in addition to the MAE. This will not be hard to do in the camera ready.
>
> **Q5**
>
> **Extensions to other activation functions** is certainly a worthy future direction, as we mentioned in Section 6. However, this is beyond the scope of the current work. The main barrier is deriving expressions analogous to Algorithm S.1 for forming the matrix $\mathbf{Q}_s$. The main reason we used the sign function is (a) it is easy to compare with existing results of Der and Lee (2005) and Neal (1996), and (b) the separating hyperplanes resulting from this activation are intuitive to understand. But convergence to a stable process occurs with other bounded activations, such as tanh, and it is not a crucial aspect of our work. The application of the stable CLT by Gnedenko and Kolmogorov only requires that the activation function is bounded. Thus ReLU might require more work, since it is not bounded. However, a detailed study of this conjecture requires computing the analogous
>  term with other activations, and is beyond the scope of the current paper.
>
>  **Extension to more than one hidden layer** is likely more tractable following techniques analogous to the *deep GP* literature (e.g., Garriga-Alonso et al., 2019, ICLR; de G. Matthews et al., 2018, ICLR). We hope other researchers can build in these directions using the platform we have laid.

---

### Official Review · Reviewer_QinA · 2024-03-27

**Q2-1 Originality-Novelty:** 2
**Q2-2 Correctness-Technical Quality:** 3
**Q2-5 Clarity Of Writing:** 2

**Q1 Summary And Contributions:**

Given the well-known result of Neal (1996), where an infinite width Bayesian neural network with one hidden layer is equivalent to a Gaussian process, the paper extends the results of Der and Lee, (2005) on unbounded variance weights to obtain an explicit characterisation of the covariance function, the conditional posterior and the posterior predictive density. This basically applies to the cases where weights are assumed to follow a symmetric $\alpha$-stable distribution. In the end, this makes the output of the neural net to converge into a $\alpha$-stable process, what allows the authors to develop their analysis and numerical experiments for values of $\alpha$ in the range $(0,2]$.

**Q2-3 Extent To Which Claims Are Supported By Evidence:**

3: Good: the main claims are supported by convincing evidence (in the form of adequate experimental evaluation, proofs, (pseudo-)code, references, assumptions).

**Q2-4 Reproducibility:**

2: Fair: key resources (e.g. proofs, code, data) are unavailable but key details (e.g. proof sketches, experimental setup) are sufficiently well-described for an expert to confidently reproduce the main results.

**Q3 Main Strengths:**

- The work is technically strong, with a particularly good review of the theoretical developments around the connection of BNNs and GPs, since the apparition of the fundamental contribution in Neal (1996).

- Even if, due to the theoretical content of the work, the clarity is at some points of the paper not very high, the authors do a great work remarking their contributions and what previous results have been already proposed in the literature. It is generally difficult to find submissions to these venues that do this sort of literature review that well.

- I find the results in Section 2.2. particularly interesting, even if they are follow-ups or extensions of the previous results of Der and Lee (2005). I cannot really evaluate if these ones are entirely novel, or easy to derive from the previously referred paper. However, I guess it is not trivial.

- The result in Corollary 1, around the deterministic and stochastic conditions in the stable and GP limits are great. Perhaps a bit more of discussion and comments on the points of the degenerate point mass would've been great.

**Q4 Main Weakness:**

- Sections 2.0 and 2.1 are extremely difficult to follow, and the context does not really fit in the rest of the paper. For instance, I couldn't follow what is $t$ and $\mathbf{t}$ there, as it is merely introduced in pp.2 after Sec 2.1. in a non-clear way. Later in Algorithm 1, a similar $t$ is used for indexing iterations, which is somehow confusing.

- Eq. (4) and Sec. 2.1. are not well prepared for being understood by the reader and do not really provide any critical info for the results in my opinion.. So they damage a bit the clarity of the paper.

- At some points of the paper, things are explained in an overcomplicated way, at least of a ML expert reader. For instance, in Sec. 2.2. the simple observation model where one has additive noise $y = f(x) + \epsilon$ is defined as the *nugget* effect coming from spatial statistics. In my opinion, these and other re-definitions of simple models are kind of unnecessary.

- I don't understand very well the impact of the result in Corollary 2. So from a Gaussian perspective when $\alpha=2$, it is saying that the distribution of the variance terms in the diagonal of the covariance matrix it is of a certain form. How this affect the main result and the ML task that the NN does?

- I'm kind of surprised by the result in Eq. (6), where we have a probabilistic integration over the posterior distribution of the covariance matrix. Usually, for the GP case, we do this wrt to the function evaluations or let's say the 'process'. What happens when $\alpha=2$ and everything reduces back to the Gaussian case? Are things still in order with that integration?

**Q5 Detailed Comments To The Authors:**

I added some comments together with my previous comments on strengths and weaknesses that could be interesting for the authors to answer.

**Q9 Complying With Reviewing Instructions:**

Yes

---

> ### Author Rebuttal · Authors · 2024-04-04
>
> [We are adding some slight edits to our rebuttal on Q3, as we noticed the referee left some further comments under Q3 (Main Strengths) as well, which merit some discussion.] We thank the referee for constructive comments. We find most of your concerns addressable and hope you will consider updating your score. We also note that our works are not merely extensions of Der and Lee (2005), as they did not consider *posterior inference.* This is the key contribution of the current work, and to our knowledge, the first work to consider this for neural networks with *even for one hidden layer*, and under *any type of activation functions*. The fact that the matrix $\mathbf{Q}_s$ is random, also hints at possibilities for learning the posterior of this kernel in a data-dependent way. We hope subsequent works can build on the platform we have laid. Responses to specific questions follow.
>
> **Q3**
>
> (Discussion on the implication of degenerate point mass). A key result, established in Corollary 1, is that the kernel $\mathbf{Q}_s$ is stochastic for $\alpha<2$ and converges to a deterministic point mass when $\alpha\to 2$. A most interesting possibility is "representation learning" under the non-GP stable limit, which is not possible under a GP limit. The key reason for this is summarized in Sec 4.2 of https://arxiv.org/abs/2108.13097, for example. Since posterior $\propto$ likelihood $\times$ prior, the posterior is non-degenerate if and only if *both* the prior and likelihood are non-degenerate. Thus, no matter what is observed, the posterior of the kernel is also a degenerate point mass at the same point for GP limit. However, for $\alpha<2$, the term $\mathbf{Q}_s$  ``retains stochasticity" even in infinite-width limit, suggesting learning a non-degenerate data-dependent posterior, aiding representation learning. This is a key feature distinguishing the current work from GP works.
>
>
> **Q4**
>
> - We will update Sec. 2.0 and 2.1 to make them clearer. The parts after Proposition 1  till the end of Section 2.1 are details that can be moved to the supplement to help the flow.  We will also add a section in the supplement providing additional background on $\alpha$-stable random variables. Here $\mathbf{t}= (t_1,\ldots, t\_n)$ denotes arguments of the characteristic function (chf) in vector and scalar forms. We will define this properly. A different index than $t$ will be used in Algorithm 1 to denote MCMC iteration.
>
> - Eq. (4) sets up the chf. This will be moved to the supplement.
>
> - It is indeed just additive Gaussian noise. We will remove the term  ``nugget effect.''
>
> - Since it has been established that $\mathbf{Q}_s$ is a random matrix, the distribution of its terms is a natural question. This is specified in Corollary 2.
>
> - We established that $\mathbf{Q}_s$ is stochastic when $\alpha < 2$. When $\alpha\to 2$, we have
> this converging to a deterministic limiting kernel. Everything is still integrable.

---

### Meta-Review · Area_Chair_o3bH · 2024-04-15

The paper proposes a methodology for posterior inference in infinite width one-layer Bayesian Neural Networks where the weights come from a prior with unbounded variance, focusing on symmetric α-stable distributions. The resulting predictive posterior is non-Gaussian providing a function space with additional properties, while it is approximated using a computationally feasible MCMC scheme. The paper is well-written, motivates well the method, and presents in an accessible manner the theoretical results and the related work. In addition, the technical part seems to be rigorous and the results are encouraging. There are some concerns, which I think the replies of the authors address and the manuscript can be updated accordingly.